# Biosynthesis of fragin is controlled by a novel quorum sensing signal

Christian Jenul[1], Simon Sieber[2,3], Christophe Daeppen[2,3], Anugraha Mathew[1], Martina Lardi[1], Gabriella Pessi [1], Dominic Hoepfner [4], Markus Neuburger[3], Anthony Linden [2], Karl Gademann [2] & Leo Eberl [1]

Members of the diazeniumdiolate class of natural compounds show potential for drug development because of their antifungal, antibacterial, antiviral, and antitumor activities. Yet, their biosynthesis has remained elusive to date. Here, we identify a gene cluster directing the biosynthesis of the diazeniumdiolate compound fragin in *Burkholderia cenocepacia* H111. We provide evidence that fragin is a metallophore and that metal chelation is the molecular basis of its antifungal activity. A subset of the fragin biosynthetic genes is involved in the synthesis of a previously undescribed cell-to-cell signal molecule, valdiazen. RNA-Seq analyses reveal that valdiazen controls fragin biosynthesis and affects the expression of more than 100 genes. Homologs of the valdiazen biosynthesis genes are found in various bacteria, suggesting that valdiazen-like compounds may constitute a new class of signal molecules. We use structural information, in silico prediction of enzymatic functions and biochemical data to propose a biosynthesis route for fragin and valdiazen.

[1] Department of Plant and Microbial Biology, University of Zurich, 8008 Zurich, Switzerland. [2] Department of Chemistry, University of Zurich, 8057 Zurich, Switzerland. [3] Department of Chemistry, University of Basel, 4056 Basel, Switzerland. [4] Novartis Institutes for BioMedical Research, Novartis Campus, 4056 Basel, Switzerland. These authors contributed equally: Simon Sieber, Christophe Daeppen. Correspondence and requests for materials should be addressed to K.G. (email: karl.gademann@uzh.ch) or to L.E. (email: leberl@botinst.uzh.ch)

Members of the *Burkholderia cepacia* complex (Bcc), a group of 20 closely related species[1], are important opportunistic human pathogens, but are also capable of synthesizing various bioactive secondary metabolites that suppress fungal and bacterial pathogens[2,3]. Many of these bioactive secondary metabolites are synthesized by multi-enzyme machineries, such as polyketide synthases (PKS) and non-ribosomal peptide synthetases (NRPS)[4,5], which are often encoded in large gene clusters together with tailoring enzymes, regulators, and transporters[6,7].

Previous work has shown that in many *Burkholderia* strains the production of antibiotic secondary metabolites is regulated by *N*-acyl homoserine lactone (AHL)-dependent quorum sensing (QS) systems. The CepIR QS system, which is present in all Bcc species, controls the production of diverse antifungal compounds, including pyrrolnitrin in *Burkholderia lata*, *B. ambifaria*, and *B. pyrrocinia*[8], and enacyloxin and an occidiofungin-like cluster also found in *B. ambifaria*[9,10]. Antifungal activity of many Bcc strains is also dependent on the presence of megaplasmid pC3, the third replicon present in all Bcc species[11]. This replicon was shown to carry the biosynthesis gene clusters directing the production of occidiofungin[12], enacyloxin[9], and pyrrolnitrin[8]. Moreover, the *afc* gene cluster, which encodes a lipopeptide antibiotic of unknown structure[13,], is present on pC3 in *B. pyrrocinia*, *B. lata* 383, *B. ambifaria* AMMD, as well as many sequenced *B. cenocepacia* strains[11]. The strain used in this study, *B. cenocepacia* H111, exhibits strong antifungal activity that is dependent on the presence of pC3[11,14] and is under the control of the CepIR QS system[8]. A bioinformatic analysis revealed that this strain does not harbor any known antifungal determinants, except for the *afc* cluster on pC3. However, in contrast to other Bcc strains, the *afc* cluster in H111 does not contribute to antifungal activity of the strain[15]. Hence, neither the genes directing the biosynthesis of the antifungal agent produced by H111 nor its structure are known.

In this study, we demonstrate that the major antifungal compound produced by *B. cenocepcia* H111 is the unusual diazoniumdiolate compound (−)-fragin (**1**). We identify the genes directing the biosynthesis of this natural compound and, on the basis of structural information, in silico prediction of enzymatic functions and biochemical data, we propose a model for its biosynthesis. We also show that a subset of the genes for fragin biosynthesis is responsible for the production of a novel signal molecule, valdiazen (**2**), which not only positively autoregulates its own and fragin biosynthesis, but is a global regulator of more than 100 genes in *B. cenocepacia* H111. Valdiazen appears to be the first member of a new class of signal molecules, as valdiazen biosynthesis gene homologs are found in various bacteria.

## Results

**Identification of the *ham* gene cluster.** The ability of *B. cenocepacia* strain H111 to suppress fungal growth depends on the presence of the megaplasmid pC3[11,15] and an intact CepIR QS system[8]. In agreement with our previous work[8,11,15], we show that the antifungal activities of the *cepI* mutant and the pC3-null strain are markedly reduced relative to the wild type (Supplementary Fig. 1) and that inactivation of the *afcA* gene, which is part of the *afc* gene cluster located on pC3, does not affect antifungal activity of *B. cenocepacia* H111 (Supplementary Fig. 2). This suggests that, for reasons that are unknown, the *afc* cluster does not contribute to the antibiotic activity of *B. cenocepacia* H111, while it is the major antifungal determinant in *B. pyrrocinia*[13] and contributes to anitfungal activity of *B. cenocepacia* K56-2[16]. To identify candidate genes for the biosynthesis of the unknown antifungal agent produced by strain H111, we compared the transcriptomes of the QS mutant *B. cenocepacia* H111Δ*cepR* and the ΔpC3 derivative with the transcriptome of the H111 parental strain[11,17]. A non-ribosomal peptide synthetase (NRPS) gene cluster consisting of seven genes was downregulated in both the Δ*cepR* and the ΔpC3 mutant (Supplementary Table 1). This gene cluster is organized in two oppositely oriented operons, comprising five (I35_4191–I35_4195) and two genes (I35_4188 and I35_4189), which we named *hamABCDE* (H111 antifungal metabolite) and *hamFG*, respectively (Fig. 1a). The NRPS-like protein HamD consists of an adenylation (A) domain, a thiolation (T) domain, and a reductase (R) domain. HamF encodes a free standing starter condensation (C) domain. Starter C domains are a recently characterized subtype of C domains, which do not catalyze peptide bond formation, but acylate L-amino acids[18]. Protein domain analysis of the remaining five genes in the *ham* cluster suggested that they are tailoring enzymes. HamA encodes a heme-like oxygenase. HamB contains an RmlC-like cupin domain, which likely acts as an epimerase or dioxygenase. The *hamC* gene encodes a *p*-aminobenzoate *N*-oxygenase, known as AurF, which catalyzes the conversion of *p*-aminobenzoate (PABA) to *p*-nitrobenzoate (PNBA)[19]. HamE is predicted to encode a polyketide cyclase/dehydrase. The last gene in the *ham* cluster, *hamG*, encodes an aminotransferase that most likely adds an amino group to the biosynthetic product of the *ham* cluster (Fig. 1b). An

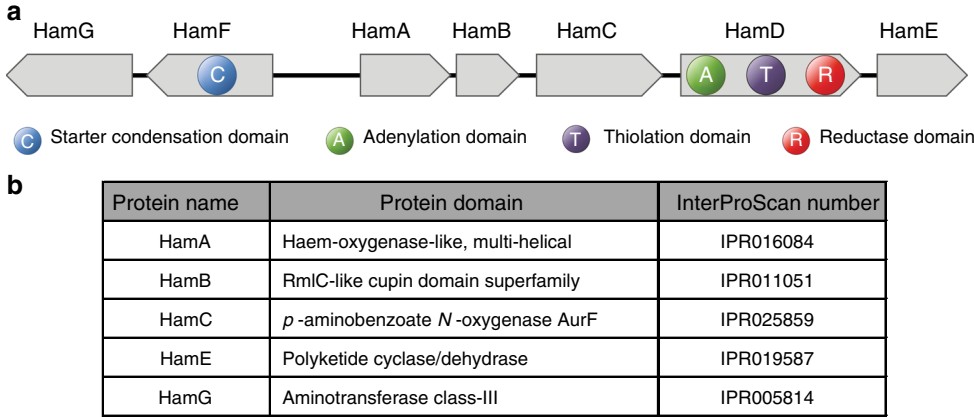

**Fig. 1** Genetic organization and protein domain predictions of the *ham* genes. **a** The *ham* cluster is organized into two divergently orientated operons. *hamD* encodes an adenylation domain, a thiolation domain and a reductase domain. *hamF* encodes a starter condensation domain. **b** Predictions of protein domains and their corresponding InterProScan numbers are shown for the five putative tailoring enzymes of the *ham* cluster

architecture search of the *ham* cluster with the MultiGeneBlast software[20] showed that homologs of the *ham* cluster are present in several *B. cenocepacia* strains and in *Burkholderia cepacia* DDS 7H-2 (Supplementary Data 1). The *hamABCDE* operon is also fully conserved in *Burkholderia plantarii* ATCC 43733, *Burkholderia glumae* PG1, *Pseudomonas aeruginosa* PA7, and *Pseudomonas fluorescens* strain X. A homologous operon that lacks *hamB* is present in *Burkholderia* sp. CCGE1003 and various *Pseudomonas* and *Pandoraea* species. The genomes of *Pandoraea vervacti* NS15 and *Pseudomonas trivialis* IHBB745 contain homologs of *hamA*, *hamC* and *hamE* and homologs of *hamC* and *hamE* are present in the genome of *Pandoraea oxalativorans* DSM23570 (Supplementary Data 1).

We also analyzed the genes in the immediate vicinity of the *ham* clusters. In eight out of nine *Burkholderia* strains, genes encoding an AraC-type regulator (CepS), a LuxR-type regulator (CepR2) and a 2-isopropylmalate dehydrogenase (LeuA) are located upstream of the *ham* cluster. In these strains, the downstream genes are also conserved and encode a LysR-type regulator and a RND-efflux pump. The *B. cenocepacia* strains and *B. cepacia* DDS 7H-2 encode a second AraC-type regulator downstream of the *ham* cluster (Supplemenatry Fig. 3a). Three genes in the downstream region of the *ham* clusters in the *Pandoraea* strains are conserved and encode a LysR-type regulator, a Fis regulator and an aldo/keto reductase. Most interestingly, a gene encoding a 2-aminoadipate aminotransferase is located immediately downstream of *hamE* and appears to be part of the *hamABCDE* operon in seven of the eight *Pandoraea* strains (Supplementary Fig. 3b). The genetic organization up- and downstream of the *ham* clusters in the *Pseudomonas* strains is highly variable. Only one gene, which codes for a GntR regulator, is often found in the vicinity of these *ham* operons (Supplementary Fig. 3c).

**The *ham* cluster directs the biosynthesis of fragin**. To investigate the importance of the *ham* cluster for antifungal activity of *B. cenocepacia* H111, in-frame deletion mutants of *hamD* and *hamF* and respective in *trans*-complemented derivatives were tested for their activity against the fungus *Fusarium solani*. Both mutants lost their antifungal activity against *F. solani*, while the complemented strains showed activities similar to that of the wild type (Fig. 2a). To ascertain the importance of individual *ham* genes for antifungal activity, we employed a *F. solani* spray assay. Inactivation of any of the seven genes of the *ham* cluster abolished antifungal activity (Fig. 2b; Supplementary Fig. 4). We noticed that in these assays the fungus produced a brownish pigment around the bacterial colonies, which may reflect a stress response of the fungus, possibly because of nutrient depletion or metabolites released by the bacteria. Expression of the wild-type *ham* alleles from a plasmid in their respective mutant background rescued antifungal activity to varying degrees (Fig. 2b; Supplementary Fig. 4). This suggests that all genes of the *ham* cluster are required for full antifungal activity.

Bioassay-guided fractionation by HPLC-MS was performed on chloroform extracts of culture supernatants of the wild-type H111 to identify the antifungal compound. A bioactive compound eluting with a retention time of 20.4 min was detected in the wild-type extract, but was missing in the extract of the *hamD* mutant (H111 Δ*hamD*) (Supplementary Fig. 5a and b). The MS spectrum of this compound displayed an *m/z* of 274.2 Da (positive mode) and an *m/z* of 272.2 Da (negative mode) suggesting a molecular mass of 273.2 Da. An additional major fragment (*m/z* of 244.2 Da, positive mode) was detected, which could originate via a loss of nitric oxide from the parent compound. The constitution of the compound was further investigated by $^1$H-NMR spectroscopy.

The $^1$H $^1$H COSY correlations between H-1 to H-2, H-2 to H-3, H-3 to H-4 and H-4 to NH were assigned to the core structure of the left part of the compound and those between H-6 to H-7, H-7 to H-(8-11) and H-(8-11) to H-12 were assigned to the core structure of the right part (Supplementary Table 2; Supplementary Note 1). The presence of a diazeniumdiolate functional group was inferred on the basis of the shift of H-3 at 4.2 ppm and C-3 at 78.0 ppm, indicating an electron-withdrawing group attached to C-3. A non-correlating de-shielded proton at 11.72 ppm, in addition to an ESI-MS fragment displaying a loss of 30 Da, constituted further evidence for the presence of a diazeniumdiolate group. These spectroscopic data are in full agreement with the structure of fragin, an antifungal compound, which was isolated from the culture supernatant of *Pseudomonas fragi* in 1967[21]. The absolute configuration of neither (+)- nor (−)-fragin has been reported in the literature, although the crystal structure of racemic fragin is known[22], and therefore we carried out an enantioselective total synthesis starting from enantiopure Fmoc-protected D- and L-valine (Fig. 3; Supplementary Note 1). First, the carboxylic acid (**3a** or **3b**) was reduced to the alcohol (**4a** or **4b**)[23], followed by a Mitsunobu reaction and subsequent reduction of the azide, which furnished the ammonium salt (**6a** or **6b**). Acylation (to give **7a** and **7b**), Fmoc-removal by AlCl$_3$[24] proceeded in excellent yields and a simple acidic and basic extraction was sufficient to obtain the primary amine (**8a** or **8b**) in pure form. The hydroxylamines (**10a** and **10b**) were obtained via a two-step procedure using dibenzoylperoxide, followed by cleavage of the benzoyl moiety in **9** in the presence of hydrazine. Isopentyl nitrite under basic conditions delivered finally enantiopure (−)-fragin (**1**) and (+)-fragin (**11**) in 83% yield (Fig. 3a). The absolute configuration of fragin was unambiguously assigned as (*R*) by comparison of the optical rotation values of synthetic and isolated compounds. X-ray crystallographic analysis of the natural product (−)-fragin (**1**) further corroborated this configuration (Fig. 4a; Supplementary Fig. 6). On the basis of quantitative HPLC, we estimated the concentration of fragin in the wild-type culture supernatant to be 69 μg/ml (253 μM).

Both fragin enantiomers showed good activity against *F. solani* (Fig. 4b), yeast and Gram-positive bacteria, but only negligible inhibition of Gram-negative bacteria (Supplementary Fig. 7a–c).

**Fragin production is positively auto-regulated**. We hypothesized that expression of the *ham* genes may be under control of a self-generated signal, as the homologous gene clusters of *Pseudomonas syringae* pv. *syringae* UMAF0158 (*mgo*) and *Pseudomonas entomophila* (*pvf*) (both of which lack *hamB* as well as the *hamFG* operon) (Supplementary Data 1) were shown previously to synthesize an extracellular signal molecule of unknown structure[25,26]. To test this possibility, we constructed a transcriptional fusion of the *hamA* promoter (P$_{hamA}$) to *lacZ* (P$_{hamA}$–*lacZ*) and introduced it into H111 wild type and H111 Δ*hamD*. The promoter activity was markedly reduced in the Δ*hamD* mutant relative to the wild-type strain when tested on agar plates (Supplementary Fig. 8a). Importantly, when the wild-type strain was streaked next to H111 Δ*hamD*/P$_{hamA}$–*lacZ*, the activity of the *hamA* promoter was restored, suggesting that the molecule is secreted and is used for cell-to-cell signaling (Fig. 5a). To ascertain the genes required for the synthesis of the signal molecule, we tested all *ham* mutants for induction of H111 Δ*hamD*/P$_{hamA}$–*lacZ* in cross-streak experiments (Fig. 5a). In full agreement with work on the *mgo* and *pvf* clusters of *P. syringae* pv. *syringae* and *P. entomophila*, respectively, we found that *hamB*, *hamF*, and *hamG* are dispensable for the synthesis of the signal. Given that the *hamFG* operon is required for fragin biosynthesis (Fig. 2b), we

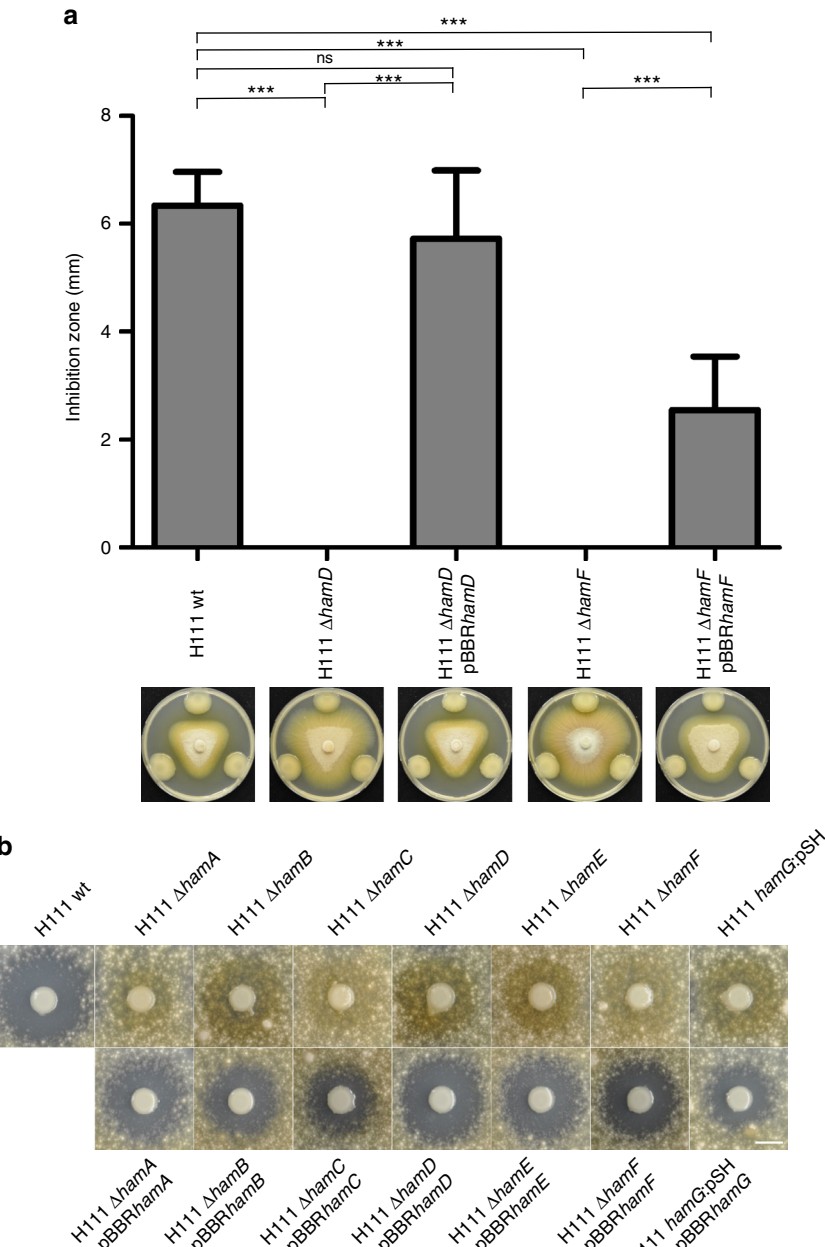

**Fig. 2** Antifungal activity is dependent on fragin, the biosynthetic product of the *ham* gene cluster. **a** Antifungal activity of the wild type (H111 wt), the *hamD* mutant (H111 Δ*hamD*), the *hamF* mutant (H111 Δ*hamF*), and the *trans*-complemented mutants (H111 Δ*hamD* pBBR*hamD* and H111 Δ*hamF* pBBR*hamF*). Results are represented as mean and error bars represent SD. Statistical analysis was performed with one-way ANOVA and Tukey's multiple comparison as post test. ns, not significant. Significance is indicated by three stars. \*\*\**p* < 0.001. *n* = 9. Representative pictures of the antifungal assays are shown below the respective bars. **b** Antifungal activity is lost in single *ham* deletion mutants and partially restored in the respective *trans*-complemented mutants. Representative pictures of antifungal spray assays against the fungus *F. solani* are shown. Scale bar indicates 10 mm

concluded that fragin cannot be responsible for induction of the *hamA* promoter, implying that the *ham* cluster directs the synthesis of two distinct molecules. We speculated that the two molecules may have similar structures and that fragin is likely to mask the presence of the signal. We therefore chose the Δ*hamF* mutant to elucidate the structure of the signal molecule. The crude extract of the Δ*hamF* mutant was purified by HPLC after a basic-acid workup and a compound, eluting with a retention time of 10.2 min, was isolated that activated *hamA* promoter activity (Supplementary Fig. 9). [1]H and [13]C NMR spectroscopy analysis revealed that the structure of the compound, which we named valdiazen (**2**) (Fig. 5b), possesses, as (−)-fragin (**1**), the

characteristic proton and carbon chemical shifts of an isopropyl and diazeniumdiolate group. Comparing the NMR spectra of the two natural products we propose that valdiazen has a hydroxy group at C4, due to a higher chemical shift (61.2 ppm instead of 39.1 ppm) (Supplementary Tables 2 and 3; Supplementary Note 2). The structure of valdiazen was confirmed by synthesis and an X-ray crystal structure analysis of synthesized (−)-valdiazen (**12**), although the latter did not independently confirm the absolute configuration (Supplementary Fig. 10). Furthermore, the presence of the two enantiomers of the natural product valdiazen was determined by chiral HPLC measurements using synthetic **12** and **13** as standards (Supplementary Fig. 11).

**Fig. 3** Enantioselective total synthesis of (−)-fragin (**1**) and (+)-fragin (**11**), and X-ray crystal structure of isolated (−)-(*R*)-fragin (**1**). Color code: blue = nitrogen; gray = carbon; red = oxygen; white = hydrogen. **a** Isobutyl chloroformiate, NaBH$_4$, DME, 1.5 h, −15 °C; **b** PPh$_3$, DEAD, DPPA, THF, 0 °C to RT, 10 h; **c** Pd/C Et$_3$SiH, MeOH/CHCl$_3$, RT, 14 h; **d** octanoyl chloride, DIPEA, DMAP, 0 °C to RT, 35 h; **e** AlCl$_3$, toluene, RT, 4 h; **f** dibenzoylperoxide, K$_2$HPO$_4$, THF, RT, 21.5 h; **g** N$_2$H$_4$ x H$_2$O, EtOH, RT, 2.5 h; **h** isopentyl nitrite, NH$_{3(g)}$, EtOH, RT, 0.5 h

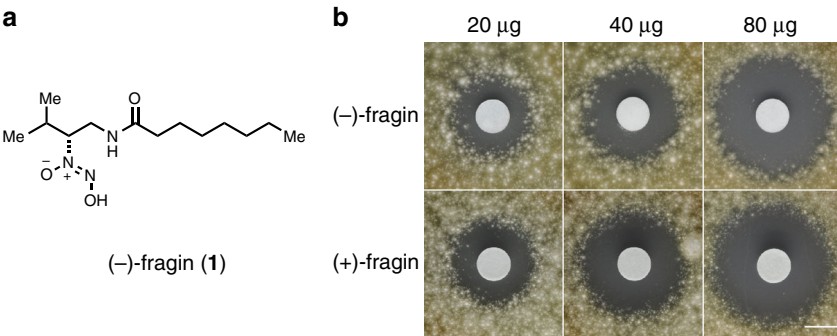

**Fig. 4** Activities of (−)-fragin (**1**) and (+)-fragin (**11**). **a** Structure of (−)-fragin (**1**). **b** Disc diffusion assays of synthetic (−)- and (+)-fragin to determine the antifungal activity. Representative pictures of an antifungal spray assay against the fungus *F. solani* are shown. Scale bar indicates 10 mm

Quantitative HPLC revealed that the concentration of valdiazen in the wild-type culture supernatant is 3.3 μg/ml (22 μM).

**Valdiazen is a novel quorum sensing signal**. To confirm that valdiazen stimulates the activity of the *hamA* promoter and to test whether the *hamFG* operon is also regulated by valdiazen, we measured β-galactosidase activities of respective transcriptional *lacZ* promoter fusions in the wild type, as well as in a Δ*hamD* (fragin and valdiazen negative) and a Δ*hamF* (fragin negative and valdiazen positive) mutant background. The activities of the two promoters were virtually indistinguishable between the wild type and the Δ*hamF* mutant but they were greatly reduced in the Δ*hamD* mutant (Fig. 5c; Supplementary Fig. 8b). We also measured the *hamA* promoter activity over the growth curve and observed that its expression is strongly induced at the late exponential phase (Supplementary Fig. 12), indicating that production of valdiazen is induced at high cell densities, which is a

typical feature of QS signals. Addition of (−)-valdiazen to the growth medium restored β-galactosidase activities in the Δ*hamD* mutant background in a dose-dependent manner, reaching the level of the wild type at a concentration of 50 μM (Fig. 5c; Supplementary Fig. 8b). These results demonstrate that valdiazen is a novel signal molecule that positively regulates its own production. In contrast, addition of (−)-fragin did not induce the *hamA* promoter (Fig. 5c). We also tested the ability of valdiazen to inhibit growth of the fungus *F. solani* and a panel of selected bacteria, but could only observe a slight inhibition of Gram-negative bacteria at the highest concentration (128 μg/ml) (Supplementary Fig. 13a and b). This shows that despite their structural similarity, fragin and valdiazen have two distinct biological functions.

To identify the valdiazen regulon, we performed RNA-Seq analyses of the wild-type strain H111 and the isogenic Δ*hamD* mutant in the absence or presence of 50 μM (−)-valdiazen. Expression of 77 genes was more than threefold reduced in the

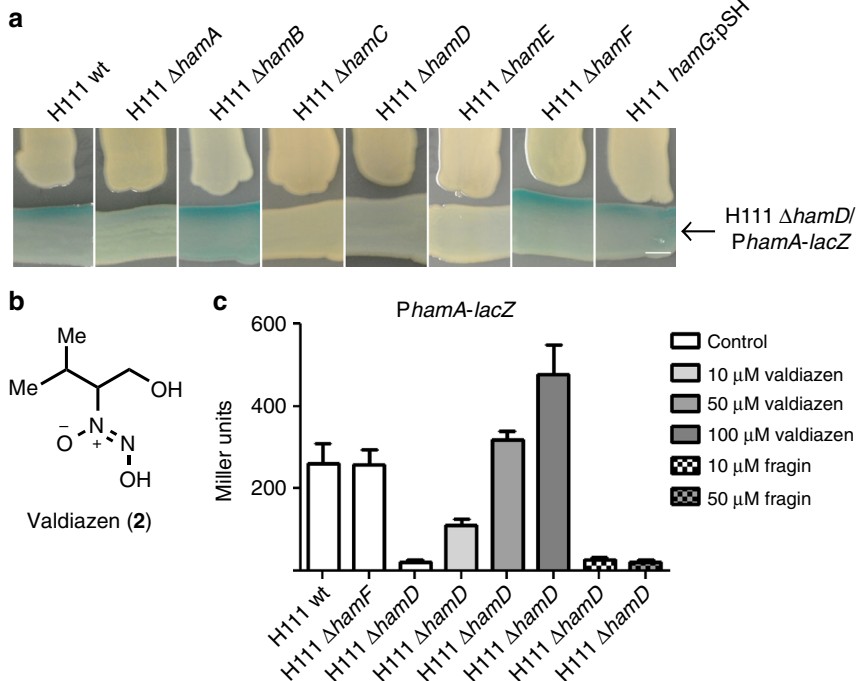

**Fig. 5** A subset of the *ham* genes synthesize the signal molecule valdiazen. **a** Cross-streaks of all seven *ham* gene mutants against the reporter strain (H111 Δ*hamD*/P*hamA-lacZ*) show that a subset of *ham* genes produces a diffusible signal molecule, which regulates *ham* promoter activity. The genes *hamB*, *hamF*, and *hamG* are dispensable for the production of the signal molecule. Scale bar indicates 5 mm. **b** Structure of valdiazen (**2**). **c** The promoter activity of the *hamABCDE* operon (P*hamA-lacZ*) was comparable in the wild-type strain (H111 wt) and the *hamF* mutant strain (H111 Δ*hamF*). The *hamD* mutant strain (H111 Δ*hamD*) showed only very low promoter activity, which could be activitated by the addition of synthetic valdiazen. Results are presented as mean and error bars represent SD. $n = 6$ (H111 Δ*hamF*, H111 Δ*hamD* 10 μM and 50 μM valdiazen), $n = 8$ (H111 wt, H111 Δ*hamD* 100 μM valdiazen), $n = 9$ (H111 Δ*hamD* 10 μM and 50 μM fragin)

H111 Δ*hamD* mutant relative to the wild type (Supplementary Data 2). Among the strongest regulated genes was a cluster of sixteen genes consisting of the two *ham* operons along with genes in their vicinity. These valdiazen-activated genes code for CepR2 (I35_4186), CepS (I35_4187), a LysR-type regulator (I35_4196), a LeuA homolog (I35_4185), a tripartite multidrug resistance system (I35_4198 – I35_4200), a hypothetical protein (I35_4201), and a TRP repeat protein (I35_4202). Valdiazen also positively regulated the expression of genes that are not physically connected to the *ham* cluster.

Of notable interest are four genes predicted to be involved in metal homeostasis, encoding a metal transporting ATPase (I35_0057), the ECF sigma factor regulating the ornibactin cluster (I35_1599)[27], a hemin uptake protein (I35_2223), and PchE (I35_6121), an NRPS enzyme involved in pyochelin biosynthesis[28]. In addition, two LeuA-like genes (I35_4185, I35_2881) and an *ilvD* homolog (I35_2532) are positively regulated by valdiazen. These three genes are involved in the metabolism of the branched chain amino acids leucine and valine.

Furthermore, several genes encoding for proteins which use metals as co-factors are also positively regulated by valdiazen (I35_2533, I35_4763, I35_4764, I35_0589). We also identified 30 genes showing significant up-regulation in the H111 Δ*hamD* mutant compared to the wild type (fold-change > 3 and *p*-value < 0.05; Supplementary Data 3). Among these are several genes involved in metal homeostasis, including the ornibactin biosynthesis cluster (I35_1607–I35_1614), the zinc regulator *zur* (I35_2669) and a zinc ABC transporter (I35_2670 – I35_2671). In addition, a hemin transport system (I35_6523–I35_6526) and two neighbor genes (I35_6527 and I35_6528) are also upregulated in the H111 Δ*hamD* mutant and therefore downregulated by valdiazen.

Comparison of the transcriptome of the wild type with the one of the Δ*hamD* mutant in the presence of 50 μM valdiazen revealed that expression of 73 genes was significantly differentially regulated (Supplementary Data 4; Supplementary Table 4). Among these, the pyochelin biosynthesis cluster (I35_1827–I35_1832) was repressed (Supplementary Data 4). These results suggest that valdiazen not only controls its own biosynthesis and the production of fragin, but also affects valine and leucine biosynthesis and regulates several genes involved in metal homeostasis.

We validated our transcriptomics data by quantitative reverse transcription PCR (qRT-PCR). Two *ham* genes (*hamC* and *hamG*) as well as two genes positively regulated by valdiazen (*ilvD* and *leuA*) and one gene negatively regulated (*hmuT*) were tested (Supplementary Table 5). All four genes showed the same regulation pattern as observed in the RNA-seq analysis.

**The antifungal activity of fragin is due to metal chelation**. We observed that the antifungal activity of *B. cenocepacia* H111 decreased with increasing amounts of iron in the test plates (Supplementary Fig. 14). In the presence of 10 μM iron, the antifungal activity of the H111 wild type and the complemented Δ*hamD* mutant was abolished. This effect was not due to downregulation of *ham* gene expression, as high iron concentrations were found to stimulate *hamA* promoter activity (Supplementary Fig. 15). This finding, together with the observation that many of the valdiazen-regulated genes are involved in metal homeostasis or encode metal-dependent enzymes, suggested that fragin may chelate metal ions and that the metal complex has no antifungal activity. When fragin extracts were spotted on antifungal test plates containing 100 μM iron, or were pre-incubated with 100 μM iron prior to spotting on plates, a

strong reduction of antifungal activity was observed (Supplementary Fig. 16a and b).

Diazenium diolate compounds are known to chelate metal ions and are best known for their ability to bind copper. Reaction of synthetic (−)-fragin with $Cu(OAc)_2$ in methanol resulted in the formation of a fragin–copper complex, which was analyzed by single-crystal X-ray crystallography (Supplementary Fig. 17). The structure analysis showed that deprotonated fragin ions ($L$) bind in a bidentate fashion to give square-planar $cis$-$Cu^{II}L_2$ complexes with crystallographic $C_2$ symmetry. The $cis$-disposition of the diazeniumdiolate ligands is probably due to electronic effects influencing their arrangement. The Cu–fragin complex no longer exhibited antifungal activity (Supplementary Fig. 16c), but still showed antibacterial activity against Gram-positive bacteria (Supplementary Fig. 16d). To further evaluate the mechanism of action, we subjected fragin to unbiased, genome-wide profiling experiments using *Saccharomyces cerevisiae* heterozygous and homozygous deletion collections[29,30]. Although heterozygous deletions (HIP) indicate targets and pathways directly affected by the compound, homozygous deletions (HOP) indicate synthetic lethality and compensating pathways to those directly affected by the compound. The HIP experiment did not identify strains with pronounced hypersensitivity against fragin (Supplementary Fig. 18a). The best scoring borderline hits were deletions in *ISD11* and *GRE1*, both annotated to be involved in regulation of iron metabolism[31,32]. In contrast to the HIP profile, the HOP experiment identified deletion strains being considerably hypersensitive (Supplementary Fig. 18b). Best scoring hits were *AFT1* and *MAC1* followed by *PEX32* and *ISU1*. *AFT1* encodes the major transcription factor involved in iron, and *MAC1* in iron and copper utilization and homeostasis[33,34]. *PEX32* encodes a peroxisomal component linked to oxidative stress, while *ISU1* is required for iron-sulfur cluster assembly in mitochondria[35,36]. The notion that the HIP profile did not outline relevant hits in combination with hypersensitive hits in HOP that are all linked to cation metabolism suggests that fragin has no direct high affinity protein target. This supports the idea that its growth-inhibitory effect is caused by its metal chelating properties, as this will result in the inactivation of enzymes that use metal ions as co-factors. This is in agreement with the abolished antifungal activity of fragin in the presence of high iron concentrations and of the synthetic Cu–fragin complex (Supplementary Fig. 16). The persistent activity of the Cu–fragin complex against Gram-positive bacteria, however, suggests that the antibacterial activity of fragin relies on a different mode of action.

**Biosynthesis of fragin and valdiazen**. On the basis of our enzyme predictions of the *ham* cluster (Fig. 1b), we speculated that HamC is most likely involved in the transformation of the amino group attached to the C3-atom of the NRPS-bound fragin/valdiazen precursor to a diazeniumdiolate group, similar to the conversion of *p*-aminobenzoic acid (PABA) to *p*-nitrobenzoic acid (PNBA) described for AurF and PsAAO, which are HamC homologs in *Streptomyces thioluteus*[19,37] and *Pseudomonas syringae* pv. *phaseolicola*[38], respectively. Indeed, purified HamC (Supplementary Fig. 19) was found to oxidize PABA to PNBA (Supplementary Fig. 20). We therefore speculated that the nitrogen group of PNBA might subsequently be transferred to the hydroxylamine precursor of fragin/valdiazen. A similar biosynthesis scheme has recently been reported for the biosynthesis of cremeomycin[39]. To test whether the hydroxylamine precursor of fragin can react with nitrite, potentially derived from PNBA or a similar molecule, to form a diazeniumdiolate, we incubated the precursor with $Na^{15}NO$ and analyzed the reaction products by

HPLC-MS. The formation of $^{15}N$-fragin along with an oxime as a major decomposition product was observed (Supplementary Fig. 21).

## Discussion

In this study, we have identified an NRPS-like cluster in *B. cenocepacia* H111, which synthesizes two distinct molecules, the antifungal molecule (−)-fragin (**1**) and a novel signal molecule, which we named valdiazen (**2**). Although fragin and valdiazen share a high degree of structural homology, their biological functions are different. Fragin was first isolated in a screen for plant growth inhibitors in 1967[21], but its biosynthesis, as well as that of any other known diazeniumdiolate natural compound, has remained elusive to date. In this study, we identified the genes required for the biosynthesis of (−)-fragin and valdiazen and, based on structural information, in silico prediction of enzymatic functions and biochemical data, we propose a putative biosynthetic route for these diazeniumdiolate compounds. In our model the amino group of L-valine, which is bound to HamD, is oxidized to a hydroxylamine by one of the tailoring enzymes, presumably the putative heme-oxygenase HamA. The *N*-monooxygenase HamC oxidizes the amino group of PABA (or of a related compound) to a nitro group (Supplementary Fig. 20), as has been demonstrated for the well-characterized homologs AurF and PsAAO[37]. We postulate that the resulting compound serves as an $NO_2$ donor to provide the distal oxidized nitrogen of the diazeniumdiolate group. A similar reaction has recently been described for the biosynthesis of cremeomycin, where L-aspartic acid is oxidized to nitrosuccinic acid by the flavin-dependent monooxygenase CreE. β-elimination of nitrosuccinic acid by the action of the CreD lyase releases nitrous acid, which then serves as the NO donor in the final diazotization step[39]. As HamE is essential for the production of both (−)-fragin and valdiazen, we hypothesize that this enzyme is also involved in the formation of the diazeniumdiolate group. However, its exact role remains to be determined. The branching point of valdiazen and (−)-fragin biosynthesis is likely the reductive release of their shared precursor from the NRPS. We suggest that, similar to the biosynthesis of myxochelin[40], the R-domain-catalyzed release of the NRPS-bound amino acid proceeds in two 2-electron reduction steps, which can competitively be transaminated during the second reduction step by a transaminase. The first R-domain catalyzed 2-electron reduction results in an aldehyde intermediate, which could undergo a second R-domain catalyzed 2-electron reduction to yield valdiazen or, alternatively, could be transaminated by the transaminase HamG. The resulting amino derivative of the latter reaction can then be acylated by HamF to yield (−)-fragin (Fig. 6). We show that HamB is essential for antifungal activity but not for the production of valdiazen. This suggests that HamB catalyzes a reaction in the biosynthesis of fragin that is downstream of the branching point of valdiazen and fragin in the proposed biosynthesis route. However, additional experiments will be required to determine its exact role in the biosynthetic pathway.

Here, we demonstrate that valdiazen, which is synthesized by a subset of the *ham* genes, is a diffusible signal that not only autoregulates its own biosynthesis and that of fragin but also affects expression of more than 100 genes (Fig. 5c, Supplementary Fig. 8b, Supplementary Data 2 and 3). We also show that the *hamA* promoter is strongly induced at the late exponential phase (Supplementary Fig. 12), indicating that production of the signal occurs at high cell densities. Taken together, valdiazen is diffusible, can induce gene expression in neighboring cells, and is produced in a population density-dependent manner. These characteristics suggest that valdiazen represents a novel QS signal.

**Fig. 6** Proposed model of fragin and valdiazen biosynthesis. The amino group of L-valine, which is bound to HamD, is transformed into a diazeniumdiolate group by the tailoring enzymes HamA, HamC, and HamE. The resulting intermediate molecule is then released in a two-step process. The first 2-electron reduction step catalyzed by the HamC R-domain results in an aldehyde intermediate, which can undergo a second 2-electron reduction to yield valdiazen. Alternatively, the aldehyde intermediate can be transaminated by HamG, resulting in an amino derivative, which can be acylated by HamF to yield fragin. A: adenylation domain, T thiolation domain, R reductase domain; e⁻ electron

Studies in *P. syringae* pv. *syringae* UMAF0158 and *P. entomophila* have identified gene clusters (*mgo* and *pvf*, respectively) that direct the biosynthesis of putative extracellular signal molecules. These gene clusters show striking similarity to the *hamABCDE* operon, but lack a *hamB* homolog, which is not required for valdiazen biosynthesis. The *mgo*-derived signal molecule was shown to regulate expression of the biosynthetic genes for mangotoxin production in *P. syringae* pv. *syringae* UMAF0158[25], whereas *pvf* was shown to be involved in regulation of virulence in *P. entomophila*[26]. Hence, valdiazen appears to be the prototype of a new class of signal molecules that is used by members of the genera *Burkholderia*, *Pseudomonas*, and *Pandorea* (Supplementary Fig. 3; Supplementary Data 1).

Among the most strongly valdiazen-regulated genes is the *ham* gene cluster and surrounding genes, including several regulators and a putative transporter system. This suggests that these genes are potentially involved in the regulation of the *ham* genes and the transport of either fragin or valdiazen. Further studies will be required to clarify their involvement in fragin and valdiazen production. The structures of fragin and valdiazen suggest valine as their precursor. In this context it is interesting to note that our transcriptomics data show a strong positive regulation of three genes involved in the biosynthesis of valin and leucine (I35_2532, I35_2881, and I35_4185). This suggests that valdiazen regulates the production of amino acids that are likely precursors in the biosynthesis of fragin and valdiazen. Valdiazen also regulates expression of genes involved in metal homeostasis, including the ornibactin biosynthesis cluster, a putative hemin uptake system and a zinc uptake system with its associated regulator. Expression of ~40% of the valdiazen-regulated genes could not be complemented by the external addition of synthetic valdiazen (Supplementary Data 2 and 3). The fact that the Δ*hamD* mutant

used in this experiment is unable to produce fragin, may indicate that the observed effects on the expression of metal homeostasis genes are indirect and are caused by the metallophore activity of fragin.

Natural products bearing a diazeniumdiolate group are rarely found in nature, but all of them show potent biological activities, including antifungal, antibacterial, antiviral and antitumor activities (reviewed in refs. [41,42]), which makes this molecule family a valuable source for drug development. Alanosine was found to chelate copper and zinc[43] and the synthetic diazeniumdiolate cupferron is commonly used as a copper chelator[44]. Interestingly, mushroom tyrosinase, which uses copper in its active center[45], is inhibited by dopastin and cupferron[46–48] and dopamin beta-hydroxylase, another copper-dependent enzyme[49], is also inhibited by dopastin[50], whereas 5-lipoxygenase, which has iron in its active center[51], is inhibited by nitrosoxacins A, B, and C[52]. Here we show that synthetic (−)-fragin chelates copper with high affinity and that the resulting fragin-metal complex has no antifungal activity. Likewise, the antifungal activity of fragin is abolished in the presence of high iron concentrations. Together with the results of the chemogenomic profiling in yeast, this demonstrates that fragin is a metallophore and that metal chelation is the molecular basis of its antifungal activity. However, additional work will be required to demonstrate unequivocally the mode of action against other organisms.

## Methods

**Bacterial strains, plasmids and growth conditions**. Strains and plasmids used in this study are listed in Supplementary Table 6. Primers used in this study are listed in Supplementary Data 5. All bacteria were routinely grown at 37 °C in broth (LB; BD™Difco, USA). For extraction of fragin and/or valdiazen, as well as promoter

activity measurements, strains were grown in ABG minimal medium (component A: 16 g $(NH_4)_2SO_4$, 48 g $Na_2HPO_4$, 24 g $KH_2PO_4$, 24 g NaCl in 800 ml $dH_2O$; component B: 2 ml 1 M $MgCl_2 \times 6 H_2O$, 0.2 ml 500 mM $CaCl_2 \times 2 H_2O$, 0.3 ml, 10 mM $FeCl_3 \times 6 H_2O$ in 800 ml $dH_2O$; component G: 20% glycerol v/v in $dH_2O$; ABG: 100 ml component A, 800 ml component B and 100 ml component G). Yeast extract (0.005%) was routinely added to boost growth except when the effect of iron on the activity of the *ham* promoter was tested. *S. cerevisae* BY4741 was grown in Yeast Peptone Dextrose (YPD) broth (BD$^{Tm}$Difco, USA) at 30 °C. *Fusarium solani* strain DS185 was routinely grown on Malt Extract Agar (MEA) (BD$^{Tm}$Difco, USA) plates at room temperature in the dark.

When required, media were supplemented with antibiotics at the following concentrations: kanamycin (Km) at 100 µg/ml, gentamycin (Gm) at 20 µg/ml, trimethoprim (Tp) at 25 µg/ml, chloramphenicol (Cm) 20 µg/ml.

**Bacterial genetics.** *B. cenocepacia* H111 mutants were constructed using three different methods. PCR amplification was carried out using the Phusion® High-Fidelity DNA Polymerase (New England BiolabsInc). Mutation of *hamG* was accomplished by single crossover. To this end an internal fragment of *hamG* was PCR amplified using the primers listed in Supplementary Data 5 and cloned into pSHAFT2Gm. The resulting plasmid was subsequently transferred into *B. cenocepacia* H111 by tri-parental mating. Correct insertion into the chromosome was verified by PCR.

Markerless mutations were either introduced using the pGPI-SceI/pDAI system or a FRT/FLP-based system. Targeted unmarked deletions with the pGPI-SceI/pDAI system were carried out as described previously[53]. In brief, upstream and downstream regions (~1 kb in size), flanking the region targeted for deletion, were amplified using the primers listed in Supplementary Data 5. The PCR products were digested with appropriate enzymes and purified with the QIAquick PCR purification kit (Qiagen, Germany). After triple ligation of pGPI-SceI with the two homology arms upstream and downstream of the targeted deletion region, the resulting plasmid was electroporated into *E. coli* SY327 and subsequently transferred into *B. cenocepacia* strains by tri-parental mating. Correct integration of the plasmid into the host genome was verified by PCR. To enforce a second homologous recombination event, plasmid pDAI-SceI was transferred into the target strain. Double crossover mutants were verified by PCR. Mutations in *hamA*, *hamB*, and *hamF* were introduced by the aid of the pGPI-SceI/pDAI system. For targeted unmarked deletions with the FRT/FLP system, upstream and downstream regions of ~1 kb in size were amplified using the primers listed in Supplementary Data 5. The PCR products were digested with the appropriate restriction enzymes and purified with the QIAquick PCR purification kit (Qiagen, Germany). The upstream fragment was cloned into plasmid pSHAFT-FRT and electroporated into *E. coli* CC118λpir. The downstream fragment was cloned into plasmid pEX-FRT and electroporated into *E. coli* MC1061. Both resulting plasmids were sequentially transferred into the target strain by conjugation. Integration of both plasmids into the chromosome via homologous recombination was verified by PCR. Plasmid pBBR5-FLP was introduced into the recipient and deletion of the targeted DNA region was verified by PCR. Mutants carrying the deletion were subsequently incubated on M9 minimal medium agar plates containing 10% sucrose as sole carbon source to cure the strain from plasmid pBBR5-FLP. Mutations in *hamC*, *hamD*, *hamE*, and *afcA* were generated by the aid of the FRT/FLP system.

**In silico methods.** Protein domains were analyzed with the NRPS/PKS analysis tool[54], NaPDoS (Natural Product Domain Seeker)[55] and the InterProScan 5[56] online tool.

The architecture search mode of the MultiGeneBlast[20] software was used to identify gene clusters similar to the *ham* cluster in other bacteria. The amino acid sequences of all Ham proteins (HamA–HamG) were combined in one fasta file and used for the search against a database created from the full bacterial GenBank subdivision. All parameters of the MultiGeneBlast software were used with default settings.

**Dual-culture plate assay.** *F. solani* was prepared by placing a fungal plug in the middle of a MEA plate and incubated for 8 days at room temperature in the dark. Bacteria from overnight LB cultures were pelleted, washed and adjusted to an $OD_{600}$ of 1. Three 10 µl samples of the bacterial suspension were spotted on MEA plates equidistant from the center and the plates were allowed to dry for 10 min. Following overnight incubation at 37 °C, plugs from the edge of 8 day old *F. solani* plates were taken and placed in the center of the MEA plates with the spotted bacteria. The plates were sealed with parafilm and incubated for 8 days at room temperature in the dark. Pictures were taken with a Nikon D90 camera with AF-S Micro Nikkor 60 mm objective and used to determine the antifungal activity of each bacterial strain. The distance between bacterial colonies and the fungus (mm) was calculated from the pictures using the ImageJ software. Chemical complementation of H111 Δ*cepI* was achieved by supplementing MEA plates with 200 nM C8-homoserine lactone (Fluka, Buchs, Switzerland).

**Fungal spray assay.** A single plug from an *F. solani* plate was incubated in a shaking flask containing 100 ml LB with 70 rpm shaking at room temperature for 2 to 3 days. The liquid *F. solani* culture was homogenized with glass beads by thorough vortexing. Bacterial strains were grown overnight at 37 °C with 220 rpm agitation, pelleted and resuspended in LB ($OD_{600}$ of 1.0) and 20 µl samples were spotted on MEA plates. After incubation at 37 °C for 20 h, the homogenized fungus was sprayed on the MEA plates. The plates were sealed with parafilm and incubated in the dark at room temperature for 48 h. The antifungal activity of the bacterial strains resulted in a fungus-free halo around the bacterial colony. Pictures were taken with a Nikon D90 camera with an AF-S Micro Nikkor 60 mm objective. Antifungal activity was defined as areas (mm²) with no fungal growth surrounding bacterial colonies and was calculated from pictures with the ImageJ software.

**Assessment of promoter activity in liquid culture.** Promoter activity of transcriptional *lacZ* fusions in liquid cultures was assessed by β-galactosidase assays as described before[57] with minor modifications. Briefly, bacterial cells were grown overnight in ABG minimal medium. Where indicated, synthetic valdiazen or (−)-fragin dissolved in methanol was added to the cultures. To assess the influence of iron on promoter activity, no yeast extract was added to the ABG medium and the $FeCl_3$ was replaced with the concentrations indicated in the results section. Bacterial overnight cultures were routinely adjusted to an $OD_{600}$ of 2, centrifuged at 5000 rpm for 5 min and resuspended in 2 ml Z-buffer. A total of 1 ml bacterial suspension was used to determine the exact $OD_{600}$ value. To permeabilize the cell membrane, 25 µl of chloroform and 0.1% SDS were added to the residual 1 ml of bacterial suspension, vortexed for 10 s and incubated at 28 °C for 10 min. Overall, 200 µl of *o*-nitrophenyl-β-D-galactoside (ONPG) solution (4 mg/ml in Z-buffer) were added to each sample, vortexed briefly and incubated at room temperature. The reaction was stopped by the addition of 500 µl 1 M $Na_2CO_3$. The samples were centrifuged at 16,000 rpm for 10 min and 1 ml of cell debris-free supernatant was used to measure the absorbance at 420 nm and 550 nm. Specific activities (Miller Units) are shown. To validate the cell density-dependent production of valdiazen, growth ($OD_{600}$) and promoter activity (Miller Units) was monitored throughout the growth curve from samples collected every 2 h for a period of 28 h.

**RNA-Seq and data analysis.** For RNA extraction, 50 ml ABG aliquots were inoculated from starter cultures (5 ml LB broth) with an $OD_{600}$ of 0.03 and incubated at 37 °C with agitation (220 rpm). Synthetic valdiazen resuspended in methanol was added from the start to a final concentration of 50 µM when appropriate. An equal volume of methanol was added to all cultures not containing synthetic valdiazen. The final methanol concentration in all cultures was 0.1%. Three independent cultures were grown to an $OD_{600}$ of 0.9 to 1 and harvested, using 1/10 of stop solution (10% phenol buffered with 10 mM Tris-HCl pH 8). RNA extraction and genomic DNA removal were performed as previously described[58]. A total of 150 ng good quality RNA (RNA integrity factor > 6) were further processed (cDNA synthesis and library preparation) using the Ovation® Complete Prokaryotic RNA-Seq Library System from NuGEN. After quantification of the obtained cDNA libraries[59], Illumina single-end sequencing was performed on a HiSeq2500 Instrument. The CLC Genomics Workbench v7.0 (QIAGEN CLC bio, Aarhus, Denmark) was used to map the sequencing reads to the *B. cenocepacia* H111 reference sequence[60]. Statistics and differential analysis was done using the DESeq software[61].

**Extraction of fragin and valdiazen.** Bacterial strains were grown in ABG minimal medium containing 0.005% yeast extract for 72 h with agitation (220 rpm) at 37 °C. Bacterial cultures were centrifuged at 5000 rpm with an Eppendorf Centrifuge 5804R. To remove all bacterial cells, supernatants were subsequently filtered with a Millipore Express$^{Tm}$ Plus 0.22 µm system. For fragin extraction, the cell free supernatant was extracted twice with 0.5 equivalents v/v chloroform in a separation funnel. The two chloroform phases were combined, dried with anhydrous magnesium sulfate (Sigma-Aldrich, Switzerland), and filtered through folded filters (grade: 3 hw; Sartorius, Switzerland). The chloroform extracts were stored at 4 °C. For valdiazen extraction, cell free supernatant was alkalized to a pH of 11 with 10 M NaOH and extracted twice with 0.5 volumes dichloromethane. The dichloromethane phases were discarded and the water phase was subsequently acidified to pH 5 with 10 M HCl and extracted twice with 0.5 volumes dichloromethane. The two dichloromethane phases were combined, dried using anhydrous magnesium sulfate (Sigma-Aldrich, Switzerland) and filtered. The dichloromethane extracts were stored at 4 °C. Chloroform and dichloromethane extracts were dried using an Eppendorf Concentrator 5301 under vacuum and routinely dissolved in 0.01 volumes methanol. The concentrated extracts were stored at −20 °C.

**Analytical methods.** All reactions were performed in dry solvents under an argon atmosphere unless stated otherwise. The anhydrous solvents were obtained from commercial suppliers and used without any purification other than that they were filtered and passed through activated anhydrous alumina columns (Innovative Technology solvent purification system) prior to use. Syringes and stainless steel cannula were used to transfer air and moisture sensitive liquids and solutions. Analytical thin layer chromatography (Merck silica gel 60 $F_{254}$ plates) was used to monitor reactions and the compounds were detected by UV light (254 nm and 350 nm) and by staining using ceric ammonium molybdate (CAM) solution followed by gentle heating with a heat gun. Flash chromatography was performed using SiliCycle silica gel 60 (230–400 Mesh) and $R_f$ values of compounds are indicated.

NMR experiments were performed at 25 °C on a Varian Gemini Bruker DPX operating at 400 MHz, on a Bruker Avance III NMR spectrometer operating at 400 MHz proton frequency, a Bruker Avance III NMR spectrometer operating at 500 MHz proton frequency and at 126 MHz carbon frequency, a Bruker Avance III NMR spectrometer operating at 600.13 MHz proton frequency and $^{1}$H-decoupling on a Varian Gemini 101 MHz spectrometer. The spectra were calibrated using the residual solvent proton and carbon signals ($\delta_H$ 7.26, $\delta_C$ 77.16 for CDCl$_3$, and $\delta_H$ 3.31, $\delta_C$ 49.00 for CD$_3$OD). Melting points (Mp) were determined using a Büchi B-545 apparatus in open capillaries and are uncorrected. IR spectra were recorded on a Varian 800 FT-IR ATR spectrometer or on a Spectrum Two (UATR) FT-IR Spectrometer (Perkin Elmer) and data are reported in terms of frequency of absorption ($v$, per cm). Optical rotations were recorded on a Jasco P-2000 digital polarimeter with a path length of 1 dm, using the 589.3 nm D-line of sodium. Concentrations ($c$) are quoted in g/100 ml. High-resolution masses (HRMS-ESI) were recorded on a Bruker max is 4G QTOF ESI mass spectrometer or a QExactive instrument (Thermo Fisher Scientific, Bremen, Germany) equipped with a heated electrospray (ESI) ionization source. HPLC purifications were performed on a UHPLC Dionex Ultimate 3000 system equipped with an Ultimate 3000 pump, an Ultimate 3000 autosampler, an Ultimate 3000 thermo-stated column compartment and an Ultimate 3000 photodiode array detector or a HPLC Agilent series 1100 equipped with a Quaternary pump, a degasser, an autosampler, a thermostated column compartment and a VWD UV detector. HPLC-MS analyses were performed on a Dionex HPLC system equipped with a P680 pump, an ASI-100 automated sample injector, a TCC-100 thermostated column compartment, a PDA-100 photodiode array detector and a MSQ-ESI mass spectrometric detector. Lyophilization was performed using a Christ Alpha 1–2 LD plus system.

X-ray crystal-structure analysis of the fragin–copper complex from the University of Basel: data collection was performed at −150 °C using CuKα radiation on a Bruker Kappa APEX diffractometer. Integration of the frames and data reduction was carried out using the APEX2 software[62]. The structure was solved by direct methods using SIR92[63]. All non-hydrogen atoms were refined anisotropically by full-matrix least-squares on $F$ using CRYSTALS[64].

X-ray crystal-structure analyses of (1) and (12) from the Univeristy of Zurich: all measurements were made at −113 °C on an Agilent Technologies SuperNova CCD area-detector diffractometer[65] using CuKα radiation. Integration of the frames and data reduction was carried out using CrysAlisPro[65]. The structures were solved by direct methods using SHELXS-2013[66]. All non-hydrogen atoms were refined anisotropically by full-matrix least-squares on $F^2$ using SHELXL-2014[67].

**Isolation of (−)-fragin (1).** A comparison of the HPLC spectra of concentrated extracts of *B. cenocepacia* H111 wild type and the *hamD* mutant identified a compound, (−)-fragin (1), that is only present in the wild-type extract. The HPLC analysis was performed using a reversed phase column (Phenomenex Gemini-NX 5 μm; 250 × 4.6 mm with flow of 1 ml/min). The column was equilibrated for 5 min with 20% MeCN/H$_2$O and the MeCN/H$_2$O applied gradient was changed from 2 to 100% MeCN in 25 min. The column was then washed with MeCN for 7 min. A peak at 20.4 min was identified as (−)-fragin (1). The separation was achieved by HPLC using a reversed phase column (Phenomenex Gemini-NX 5 μm; 250 × 4.6 mm with flow of 1 ml/min). The column was equilibrated for 5 min with 20% MeCN/H$_2$O and the MeCN/H$_2$O applied gradient was changed from 20 to 80% and finally to 100% MeCN in 15 and 1 min, respectively. The column was then washed with MeCN for 8 min. A peak at 14.8 min was isolated and the antifungal activity of this fraction was tested. The procedure was repeated using multiple HPLC runs with the same conditions described above and (−)-fragin (1.4 mg) was obtained as a white solid. NMR experiments were performed in CDCl$_3$ that was filtered through aluminum oxide (activated, basic). (−)-Fragin (1.4 mg) was dissolved in warm toluene (40 °C) and cooled slowly to RT to obtain a single crystal, which was analyzed by X-ray crystallography. The absolute configuration of (−)-(R)-fragin (1) was determined by the crystal structure analysis and by comparing the optical rotation values of the natural and synthetic compound.

(−)-(R)-Fragin (1): white solid; MS (ESI): [M+H]$^{+}$: 244.2 and 274.2; [M−H]$^{−}$: 272.2; Optical rotation: [$\alpha$]$_D$ −122° (c1.9, EtOH); $^{1}$H-NMR: (500 MHz, CDCl$_3$) $\delta$ = 11.72 (s, 1H), 5.69 (s, 1H), 4.20 (td, $J$ = 9.2, 3.1 Hz, 1H), 3.86 (ddd, $J$ = 14.4, 6.0, 3.1 Hz, 1H), 3.60 (ddd, $J$ = 14.4, 9.4, 6.1 Hz, 1H), 2.25–2.17 (m, 1H), 2.14 (td, $J$ = 7.4, 1.8 Hz, 2H), 1.61–152 (m, 2H), 1.31–1.23 (m, 8H), 1.07 (d, $J$ = 6.8 Hz, 3H), 0.90 (d, $J$ = 6.7 Hz, 3H), 0.87 (t, $J$ = 6.8 Hz, 3H); $^{13}$C NMR: (from HMQC, CDCl$_3$) $\delta$ = 173.7, 78.0, 39.1, 36.6, 31.8, 29.3, 29.1, 29.1, 25.7, 22.7, 19.1, 18.9, 14.2 (as determined from the $^{13}$C NMR of the synthetic compound).

**Reactivity of the hydroxylamine 10a with sodium nitrite.** To a solution of the hydroxylamine 10a (1.4 μM, 0.5 ml) in MeOH, Na$^{15}$NO$_2$ (6.9 mg) and water (0.5 ml) were added. The solution was filtered and analyzed by HPLC-MS (ESI positive mode). Three data points were recorded: after the preparation of the sample, after 24 h and after 44 h.

**Isolation of valdiazen (2).** Concentrated valdiazen extracts were separated using multiple RP-HPLC runs (Phenomenex Synergi Hydro-RP 4 μm; 250 × 4.6 mm, 40 °

C) at a flow of 1 ml/min. The MeCN/aq. NH$_4$OAc (20 mM, pH = 4) gradient applied was changed from 2 to 50% and finally 50 to 100% in 15 and 7.1 min, respectively. The column was then washed with MeCN for 4 min. Valdiazen, which has a retention time of 10.2 min, was collected during each run. The fractions were combined, rotary evaporated, dissolved in CH$_2$Cl$_2$ (5 ml), filtered over wool and evaporated to dryness to afford 0.5 mg of valdiazen (2) as a white solid.

Valdiazen (2): white solid; $^{1}$H-NMR (500 MHz, MeOD): $\delta$ = 4.04–3.96 (m, 2H), 3.87–3.80 (m, 1H), 2.11 (ddh, $J$ = 13.3, 7.2, 6.7, 1H), 1.02 (d, $J$ = 6.8 Hz, 3H), 0.90 (d, $J$ = 6.8 Hz, 3H); $^{13}$C-NMR: (126 MHz, MeOD) $\delta$ = 82.2, 61.2, 29.3, 19.4, 19.4.

**HPLC chiral separation of valdiazen (2).** Valdiazen (2) was separated by HPLC using a chiral column (Chiralpak OD-H 10 μm; 250 × 4.6 mm) at a flow of N0.8 ml/min. The hexane/EtOH isocratic gradient was 80%. Control samples of pure (−)-valdiazen (12) eluted at 10.3 min, while (+)-valdiazen (13) eluted at 7.6 min. Two corresponding peaks at 10 min and 7.5 min were observed for valdiazen (2).

**Transcriptional lacZ fusions.** DNA fragments containing the putative promoters of the *hamABCDE* and *hamFG* operon were amplified with Phusion® High-Fidelity DNA Polymerase (New England BioLabs Inc.) according to the manufacturer's instructions using the primers listed in Supplementary Data 5. Annealing temperatures were calculated with the NEB Tm calculator online program (New England BioLabs Inc.). The promoter fragments were separated on agarose gels, excised and purified with the QIAquick gel extraction kit (Qiagen). The purified fragments were digested with the restriction enzymes (New England BioLabs Inc.) indicated in Supplementary Data 5 according to the manufacturer's instructions and purified by the QIAquick PCR purification kit (Qiagen). The *lacZ*-fusion vector pSU11[68] was digested with the restriction enzymes XhoI and HindIII (New England BioLabs Inc.) according to the manufacturer's instructions and subsequently purified by the QIAquick PCR purification kit (Qiagen). The promoter fragments were ligated into pSU11 with T4 ligase (New England BioLabs Inc.) according to the manufacturer's instructions. The resulting plasmids were electroporated into *E. coli* Top10 and subsequently transferred into the target strain by tri-parental mating and selected on PIA plates containing gentamycin (20 μg/ml).

**Quantitative real-time PCR.** RNA was extracted from *B. cenocepacia* H111 wild type and Δ*hamD* cells grown to late exponential phase in ABG medium, as previously described[69] and further purified using the RNeasy Qiagen kit (Qiagen, Germany). First strand cDNA was synthesized using random primers (Invitrogen, USA) and MLV reverse transcriptase (Promega, USA). qPCR was performed on the generated cDNA using Brilliant III Ultra-Fast SYBR® Green QPCR Master Mix (Agilent, USA) and a Mx3000P instrument (Agilent, USA). Primers used are listed in Supplementary Data 5. Each PCR reaction was run in triplicate and melting curve data was analyzed to determine the PCR specificity. Relative expression levels of several differentially regulated genes from the RNA-seq data were calculated using the ΔΔ CT method[70] and the *rpoD* gene was used as the reference gene for normalization.

**Construction of plasmids for the complementation of mutants.** Plasmids for complementation purposes were constructed by PCR of the respective gene using the Phusion® High-Fidelity DNA Polymerase (New England Biolabs Inc). For all genes, except *hamB*, the amplified fragment included the predicted ribosome binding site (RBS) of the gene. The 5′-end of *hamB* overlaps with the 3′-end of *hamA* and does not have its own RBS; therefore, a synthetic RBS was introduced with the forward primer upstream of the start codon of *hamB*. All primers used for the construction of these plasmids are listed in Supplementary Data 5. The annealing temperatures for the PCRs were calculated with the NEB Tm calculator online program (New England BioLabs Inc.). The amplified fragments were separated on agarose gels, excised and purified with the QIAquick gel extraction kit (Qiagen). The purified fragments were digested with the restriction enzymes (New England BioLabs Inc.) indicated in Supplementary Data 5 according to the manufacturer's instructions and again purified by the QIAquick PCR purification kit (Qiagen). The expression vector pBBR1MCS2 was digested with the restriction enzymes HindIII and XbaI (New England BioLabs Inc.; expression of *hamA*, *hamB*, *hamD*, *hamF*, and *hamG*) or XhoI and XbaI (New England BioLabs Inc.; expression of *hamC* and *hamE*) according to the manufacturer's instructions and subsequently purified by the QIAquick PCR purification kit (Qiagen). The promoter fragments were ligated into pSU11 with T4 ligase (New England BioLabs Inc.) according to the manufacturer's instructions. The resulting plasmids were electroporated into *E. coli* Top10 and subsequently transferred into the target strain by tri-parental mating.

**Plasmid transfer.** Plasmids were transferred from *E. coli* to *B. cenocepacia* strains by tri-parental mating using *E. coli* HB101/pRK600 as the helper strain as described previously[71]. *E. coli* strains (donor and helper strain) and *B. cenocepacia* (recipient strain) were grown in LB overnight with agitation (220 rpm) at 37 °C. The donor and helper strains were mixed and incubated at room temperature for 10 min

before the recipient strain was added. The bacterial mixture was then spotted on LB agar plates and incubated at 37 °C for 4 to 7 h, washed off with saline (0.9% NaCl solution) and plated on Pseudomonas Isolation Agar (PIA; BD™Difco, USA) containing appropriate antibiotics and incubated overnight at 37 °C. Transconjugants were verified by PCR using the primers listed in Supplementary Data 5.

**HamC protein purification and enzyme assay.** The *hamC* gene was amplified with Phusion® High-Fidelity DNA Polymerase (New England BioLabs Inc.) using the primers listed in Supplementary Data 5. The resulting PCR product was purified from an agarose gel by the QIAquick gel extraction kit (Qiagen), digested with the restriction enzymes indicated in Supplementary Data 5, and cloned into pQE30. The resulting plasmid, pQEhamC (Supplementary Table 6), was electroporated into *E. coli* M15. 200 ml LB were inoculated with an overnight culture of *E. coli* M15 pQEhamC to an $OD_{600}$ of 0.02 and grown with agitation (220 rpm) at 37 °C. After 3 h of growth, the cultures were shifted to 30 °C and the expression of his-tagged HamC was induced with 500 μM IPTG. Every hour, starting at the induction, 100 μM ferrous ammonium sulfate was added to the cultures to ensure incorporation of iron into the protein. After 3 h, the cells were centrifuged (3000×*g*, 4 °C, 10 min) and the pellet was frozen in liquid nitrogen and stored at −80 °C. For protein purification, the cells were thawed on ice and resuspended in 5 ml lysis buffer (100 mM $NaH_2PO_4$; 500 mM NaCl, 5 mM Imidazol; 10% Glycerol; pH8) containing protease inhibitor (Roche Protean Tablet EDTA free) and lysed with a French press at 2 kBar. The cell lysate was centrifuged at 10,000×*g* at 4 °C for 60 min. Purification of his-tagged HamC was performed with Profinity™ IMAC Resin (BioRad) according to the manufacturer instructions with slight modifications. In brief, the lysate was incubated with washed resin for 60 min at 4 °C with shaking, centrifuged (1000×*g*, 4 °C, 2 min) and washed twice with washing buffer (50 mM $NaH_2PO_4$, 500 mM NaCl, 30 mM Imidazol, pH8). The pellet was resuspended in 500 μl elution buffer (50 mM $NaH_2PO_4$, 500 mM NaCl, 250 mM Imidazol, pH8) and centrifuged (1000×*g*, 4 °C, 2 min). The supernatant was collected, flash frozen with liquid nitrogen and stored at −80 °C. The elution step was repeated three times and the collected supernatants pooled. Washing and elution steps of his-tagged HamC were examined by SDS-PAGE and Coomassie blue staining. The eluted protein was subsequently concentrated and the buffer exchanged (25 mM MOPS, 400 mM NaCl, pH7) using Amicon Ultra Centrifugal Filter Units (NMWL 10 kDa) (EMD MILLIPORE). The purity of the concentrated protein was estimated by SDS-PAGE and the protein concentrations were determined with Bradford Reagent (Sigma-Aldrich).

**Enzyme activity.** The enzymatic activity of purified HamC was studied as previously described for PsAAO[38] with modifications. To a solution of water (76.1 μl), PABA (12.5 μl, 40 mM in DMSO), NaCl (1.8 μl, 1 M in $H_2O$), MOPS (2.3 μl, 1 M in $H_2O$) and a solution of the enzyme HamC (6.4 μl) in MOPS buffer (25 mM) with NaCl (100 mM) were sequentially added. To start the reaction $H_2O_2$ (5 μl, 30% in $H_2O$) was added and the reaction was analyzed by HPLC-MS after 1 day. A control experiment was performed using water instead of the enzyme solution.

**Chemogenomic profiling including HIP and HOP.** The growth-inhibitory potency of test substances was determined using wild type *S. cerevisiae* BY4743. $OD_{600}$ values of exponentially growing cultures in rich medium were recorded with a robotic system. Twelve-point serial dilutions were assayed in 96-well plates with a reaction volume of 150 μl, start $OD_{600}$ was 0.05. Solutions containing dimethyl sulfoxide (DMSO) were normalized to 2%. $IC_{30}$ values were calculated using logistic regression curve fits generated by TIBCO Spotfire v3.2.1 (TIBCO Software Inc.). HIP, HOP, and microarray analysis were performed as described previously[30]. Sensitivity was computed as the median absolute deviation logarithmic (MADL) score for each compound/concentration combination. *z*-scores are based on a robust parametric estimation of gene variability from over 3000 different profilings and were computed as described in detail in Hoepfner et al.[30]

**Quantification of valdiazen.** The concentration of valdiazen (**2**) in the supernatant of the H111 wild type and the H111 Δ*hamF* strain was determined by UHPLC/MS measurements (ultimate 3000 system equipped with a Kinetex® EVO C18; 1.7 μm; 100 Å, 50 × 2.1 mm as the column, a flow of 0.4 ml/min and a solvent system consisting of MeCN/$H_2O$ (both containing 0.1% formic acid)). The buffer used varied from 2 to 2% in 0.9 min, 2 to 30% in 0.3 min, 30 to 95% in 2.3 min, 95 to 100% in 0.05 min and the column was washed with 100% MeCN for 1.24 min. The samples used for the calibration curve were composed of four aqueous solutions of the synthetic valdiazen (Supplementary Methods) at 10, 20, 50, and 100 μg/ml and 1 μl was injected. The supernatants of the H111 wild type and the H111 Δ*hamF* mutant were filtered and 2 μl were injected. For the quantification, the chromatograms at wavelengths between 240 and 250 nm were analyzed and, to confirm the identity of the peak, SIM measurements at 149.1 Da were performed. A valdiazen concentration of 6.0 μg/ml (41 μM) was calculated in the supernatant of the H111 Δ*hamF* mutant and 3.3 μg/ml (22 μM) in the supernatant of the H111 wild-type strain.

**Quantification of (−)-fragin.** The quantification of fragin (**1**) in the supernatant of the H111 wild type strain was performed as described for the quantification of

valdiazen with some modifications. The solvent system of the UHPLC was composed of MeCN/$H_2O$ (both containing 0.1% formic acid), the column was equilibrated at 30% for 0.4 min, the gradient was varied from 30 to 95% in 2.1 min, 95 to 100% in 0.05 min and the column was washed with 100% MeCN for 1.24 min. The samples used for the calibration curve were composed of synthetic fragin (Supplementary Methods) solutions ($H_2O$/MeCN 1/1) from 50, 100, 200 and 500 μg/ml and 1 μl of these samples and the supernatant of H111 wild-type strain were injected. The quantification was done using the SIM chromatograms at 244.1 Da. The concentration of fragin was calculated to be 69 μg/ml (253 μM) in the supernatant of the H111 wild-type strain.

**Statistical analysis.** Statistical significance for differences of antifungal activity in dual-culture plate assays was calculated with either one-way ANOVA and Tukey's multiple comparison as a post test or a paired two-tailed Student's *t*-test. Differences were considered significant with $p < 0.05$.

**Data availability.** Crystallographic data have been deposited in the Cambridge Crystallographic Data Centre (www.ccdc.cam.ac.uk/structures) with accession codes CCDC-1543487, CCDC-1543488, and CCDC-1812456. The RNA-Seq raw and processed data have been deposited in the GEO database with accession code GSE97171. The authors declare that all other data supporting the findings of this study are available in this article and its Supplementary Information files, or from the corresponding authors upon request.

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

## Acknowledgements

We thank Isabel Scholl and Dr. Yilei Liu for technical assistance. We want to thank Prof. Grossniklaus and Dr. Krügel for gifting us the yeast strain *Saccharomyces cerevisae* BY4741. This work was supported by the Swiss National Science Foundation (Project 31003A-169307/1) to L.E. and (Project 200020_163151) to K.G.

## Author contributions

C.J., L.E., and K.G. designed the study. C.J. performed in silico analysis of the gene cluster and protein domain predictions, performed all genetic manipulations, plasmid constructions, promoter studies, antifungal, and antibacterial assays. C.J., S.S., and C.D. isolated and purified the natural compounds. C.J. performed heterologous protein expression and purification and S.S. characterized enzyme activities. S.S. and C.D. elucidated the natural product structures and performed all chemical syntheses. S.S., C.D., and K.G. analyzed chemical data. A.M., M.L., and G.P. performed RNA-seq experiments.

A.M. performed qRT-PCR, growth curves, promoter fusion studies, and antifungal assays. D.H. performed the HIP-HOP analysis. A.L. and M.N. performed X-ray crystal-structure analyses. C.J., S.S., C.D., K.G., and L.E. prepared the manuscript.

## Additional information

**Competing interests:** The authors declare no competing interests.

