## [Peer Review File(PDF 552 kb) · Nature Communications]

Reviewers' comments:

Reviewer #1 (Remarks to the Author):

GENERAL COMMENTS

The manuscript entitled "Biosynthesis of fragin is controlled by a novel quorum sensing signal", but Jenul et al., could be suitable for its publication after some major revisions. Under my point of view, this is a nice piece of work, but some issues must be addressed before having it ready for publishing.

In this manuscript, authors confirmed production of fragin by *B. cenocepacia*, but also confirmed the production of a novel chemical signal (valdiazin) that could be used for microbial communication, and also get its molecular structure. Their role has been studied, and the possible interaction with iron is proposed. Thus, the manuscript presented novel interesting results that could have an impact in other microbial systems. It is well written and concise. Experiments performed are rigorous and the results are sound. Data shown are quite clear, but no statistics have been used, which is strongly recommended. However, some issues arise about the antifungals produced by *B. cenocepacia* H111 and their interaction that must be clarified.

MAYOR CONCERNS

In the Introduction section, it is clearly stated the role in the antifungal activity of the quorum sensing regulatory system CepI/R (regulation of pyrrolnitrin, enacycloxin or occidiofungin). Moreover, it is also detailed the key role of plasmid pC3 in the antifungal phenotype, since it contains the genes for production of those antifungals. However, no information is presented about if *B. cenocepacia* H111 (which harbours pC3) produced or not those antifungals.

I was expecting the evaluation of that signal molecule in the production of the other major or main antibiotics already described. Deletion of cepIR showed decrease in fungal antagonism (supplementary Fig 1; I would like also to see included here the complemented strain with the cepI/R system restored). Moreover, the Δ pC3 derivative strains should be also included in this and other figures, in order to confirm the loss of antagonism (and to be used as negative control for antagonism or antifungal production).

If production of other antifungals is reported, additional information should be included: are these antifungals relevant for antagonism? Are their production regulated together with other antifungals? How could work the coordinated and different regulation systems? Have they an impact on antifungal production? In the case that H111 does not produce other antibiotics, what is happening with those genes present in pC3 then?

All these aspects are not clear enough to really understand the role of fragin and valdiazin, and their influence in the antagonistic phenotype. Thus, to my opinion, this aspect must be clarified and additional experimentation included, if necessary.

OTHER CONCERNS

Statistic is absent in all the figures. Clear differences can be seen, but statistic will strengthen the results.

Lines 104-105: I do not see the "abolishment" of the antifungal activity in Fig. 2b. It is clear that other antifungal compound is still being released to the medium by the different derivative strains. It is not clearly stated if there are other antifungals released or different concentrations of one or various compounds, but a halo can be more or less observed in all the derivative mutants, but especially in Δ hamB, Δ hamD, Δ hamE and Δ hamG. An explanation would be necessary.

Line 190: Change "hamD" to " Δ hamD"; change "hamF" to " Δ hamF". Same in the forthcoming pages for naming mutants.

Lines 367-369: Is fragin a siderophore? Please, add some comments about it after your claims in these lines. Also shown in Supplementary Figures 12 and 13.

Figure 1b: additional data of the protein domains of the ham genes should be included. Identity, coverage, enzyme commission numbers, etc.

Figure 2: Please, correct the column H111 Δ hamF pBBR/hamF. It is repeated twice.

Figure 5: Please, clarify the role of HamB in this figure.

Supplementary Fig. 3. A comparison of all these ham genes could be constructed, in order to infer phylogenetic and comparative purposes among those bacterial strains. Moreover, even the analysis of the two-three genes down- and upstream the ham genes could give a valuable information about similarity among strains.

Reviewer #2 (Remarks to the Author):

In this manuscript, the authors determined the absolute configurations of two structurally related molecules, an antifungal molecule fragin and a novel signal molecule valdiazin. They demonstrated the functions as well as positive biosynthesis routes of the two molecules. Valdiazin was presented to be a new signal molecule controlling the expression of more than 150 genes, including itself and fragin. The authors also suggested the possible mode of action of the two molecules.

The major concern for the manuscript is that the premise that valdiazin is a global regulator is based on only RNA seq data (without sufficient controls) and homology in other bacteria. Valdiazin was added at a 100 μ M concentration at the start of the RNA seq experiment. This concentration has to be shown to be physiologically relevant under natural conditions of the bacterium. This concentration is also high for a compound that may have membrane perturbation activity (as possibly observed in Gram positive inhibitory activity). This nonspecific activity of membrane perturbation could be attributed to the observed RNA-seq data. Additional controls are needed in the experiments. It would be useful to isolate valdiazin from one of the fragin deficient bacteria and characterize its regulatory based activity. It is not uncommon for bio-intermediates to regulate the synthesis of end products. I think the authors have demonstrated this activity sufficiently.

Additional comments:

1. In the title, valdiazin was suggested to be a quorum sensing signal. Although the authors showed that CepIR QS mutant has around 3/7 of the bioactivity of wild type in supplementary figure 1, and hamABCDE and HamFG operons were down-regulated in cepR mutant, I do not think they explained the relationship between CepIR quorum sensing system and the ham cluster clearly. The reason for valdiazin to be considered as a QS signal needs to be addressed. Furthermore, global regulator and quorum sensing is different. QS would suggest that concentration is cell density dependent and no data was presented to support this.

2. Figure 2a, the area of inhibition of Δ hamD is about 1/3 of wt strain, however, in line 111, the authors stated that no bioactive compound was detected in the extract of the hamD mutant. I was wondering why the hamD mutant is still active? Also in figure 2a, the last two labels in X axis are the same. I assume that the second last one should be H111 Δ HamF. The HamF complement strain only has 2/3 of the activity of wt, and I wouldn't think it showed similar activity with wild type (line 102). Are the differences between the strains significant?

3. Figure 2b, I am a little bit concerned about some of the complement strains. For example, the complement strains of Δ HamB, Δ HamC and Δ HamG seem to be less active than the wt, however the authors claimed that the complemented strains showed wild type activity in line 106. I think it would be worthwhile to show the area of inhibition of all complemented strains and wt for better comparison.

4. Supplementary figure 4, I was wondering if any peak other than fragin has any activity, based on the fact that Δ HamD has around 1/3 bioactivity as wild-type. Figure 4b, is the concentration of fragin comparable between crude extract and the HPLC-purified compound? What is the bioassay of the crude extract of hamD mutant looks like?

5. Figure 4c, the promoter activity of the hamABCDE operon was restored to wt level by supplementing 50 μ M valdiazin to hamD mutant. Line 206, the authors performed RNA-seq of the

wild type and the hamD mutant in the absence or presence of 100 μ M valdiazin. Some explanation for the concentration choice is needed.

6. Supplementary table 9, 3.85% of the fragin is labeled with L-leucine 13C. However, in line 301, "a 4.4% increase of 13C incorporation into fragin was observed for the leucine labeling experiment", why are the two numbers different? Line 302, the authors stated that valine is the precursor of fragin, because the incorporation of labeled valine is more than two-fold higher than that of labeled leucine. Although the incorporation rate of labeled leucine is lower than valine, the rate for leucine is not low, and leucine was predicted to be the substrate of HamD A domain (line 293). This data is not sufficient for determining substrate. First, Burkholderia are capable of utilizing just about anything as a carbon source and second, there is no data supporting that both are taken up by the bacterium at equal rates and amounts. The data as I see it, is that both are utilized by the bacterium.

7. Line 219, it is suggested that valdiazin activates genes involved in the metabolism of 3-methyl-2-oxobutanoate, which is the branching point between valine and leucine biosynthesis pathway. I am curious about how valdiazin affects the metabolism of 3-methyl-2-oxobutanoate. Is it facilitating the synthesis of one of the two amino acids, and is that involved in the self-regulation of the production of valdiazin?

8. Line 98, no data to support it being a non-polar mutant

9. Line 107, need to be careful with the statement. Synthesis and transport is also important for the bacterium's antifungal activity.

10. Line 127, discussion of the 1967 paper comes after the discussion of the structure. It is out of place and should be mentioned earlier on to put your data into perspective.

11. Line 208, there is no discussion for only half of the genes being restored.

12. Line 249 Fragin ions (spelling)

13. Line 255-271, there are no methods describing the HIP and HOP assays.

14. Line 312, try not to use words like surprising

15. Lines 435-437, More information or reference is needed.

16. Line 462, what was the final percentage of methanol?

17. Line 498, reference needed

Reviewer #3 (Remarks to the Author):

This manuscript is to report the gene cluster required for the biosynthesis of the diazenium diolate compound fragin in Burkholderia cenocepacia strain H111. The research shows that fragin, a plant growth inhibitor, is the major antifungal compound synthesized by this bacterium. Valdiazin has been characterized to be a novel cell-to-cell signaling molecule and its biosynthesis genes have been identified in this study. The valdiazin regulon includes production of itself, and fragin biosynthesis and other >150 genes. In addition, homologs of the biosynthesis genes of valdiazin were identified in various bacteria. Seems to me that the findings may bring a wide impact on understanding gene a novel class of signal molecules. Basically, the research was hypothesis driven, the most experiments were designed logically and the manuscript were prepared nicely. The findings have expanded the knowledge on functions of fragin and could make significant impacts on studies of bacterial gene regulation.

Major questions, suggestions or comments:

1) Fragin has inhibitory actions on growth of plants, fungi and bacteria, which indicates that it has a very broad range of functions. Is the fragin activity fungicidal or fungistatic? Understanding its mode of action would beef up the research significantly. In addition, poor specificity may be a hurdler for drug development.

2) For antifungal assays, suggest using fungal spore-based methods for evaluation of antifungal

activity. Typically, almost of all the strains of the *Fusarium* species easily produces conidia on common media. Spore-based method could generate more accurate quantitative results as compared agar plug. In Fig. 2b, the mutants produced different phenotypes regarding antifungal activities, which should be discussed in the manuscript. The strain names of *F. solani* (Line 387), yeast (Line 386) and bacteria (Suppl. Fig. 6) should be provided.

Minor questions, suggestions or comments:

- 1) Methods for generation of the plasmids used for genetic complementation assays should be provided as supplementary data.
- 2) All scientific names of organisms in the manuscript including in the reference list should be in italic.
- 3) How many biological replicates were conducted for RNA-seq results?

Response to reviewers' comments:

Reviewer #1 (Remarks to the Author):

GENERAL COMMENTS

The manuscript entitled "Biosynthesis of fragin is controlled by a novel quorum sensing signal", but Jenul et al., could suitable for its publication after some major revisions. Under my point of view, this is a nice piece of work, but some issues must be addressed before have it ready for publishing.

In this manuscript, authors confirmed production of fragin by *B. cenocepacia*, but also confirmed the production of a novel chemical signal (valdiazin) that could be used for microbial communication, and also get its molecular structure. Their role have been studied, and the possible interaction with iron is proposed. Thus, the manuscript presented novel interesting results that could have an impact in other microbial systems. It is well written and concise. Experiments performed are rigorous and the results are sound. Data shown are quite clear, but not statistic have been used, which is strongly recommended. However, some issues arise about the antifungals produced by *B. cenocepacia* H111 and their interaction that must be clarified.

>> We agree with the reviewer that we did not provide sufficient background on the antifungals produced by our model strain H111 and we have amended the revised manuscript accordingly. We have also performed statistical analyses and added this information.

MAYOR CONCERNS

In the Introduction section, it is clearly stated the role in the antifungal activity of the quorum sensing regulatory system *CepI/R* (regulation of pyrrolnitrin, enacycloxin or occidiofungin). Moreover, is also detailed the key role of plasmid pC3 in the antifungal phenotype, since it contains the genes for production of those antifungals. However, no information is presented about if *B. cenocepacia* H111 (which harbours pC3) produced or not those antifungals. I was expecting the evaluation of that signal molecular in the production of the other major or main antibiotics already described.

>> We want to thank the reviewer for this important comment. Previous work has shown that both quorum sensing and the presence of plasmid pC3 is important for antifungal activity of many *Burkholderia* species. We have shown that this is also true for *B. cenocepacia* H111. Members of the genus *Burkholderia* produce a wide range of antifungal compounds and many gene clusters directing their biosynthesis have been identified. However, a bioinformatics analysis revealed that H111 does not harbor any of the known antifungal determinants (including those coding for pyrrolnitrin, enacycloxin and occidiofungin synthesis), except for the *afc* cluster on pC3. However, our recent study showed that, in contrast to other Bcc strains, the *afc* cluster in H111 does not contribute to antifungal activity of the strain (Agnoli et al., AEM 83, 2017). Hence, neither the genes directing the biosynthesis of the antifungal agent produced by H111 nor its structure was known. We added this information along with the relevant citations in the introduction and hope that it clarifies why we could not evaluate the role of the signal in the production of other major or main antibiotics already described.

Deletion of *cepIR* showed decrease in fungal antagonism (supplementary Fig 1; I would like also to see included here the complemented strain with the *cepI/R* system restored). Moreover, the Δ pC3 derivative strains should be also included in this and other figures, in order to confirm the loss of antagonism (and to be used as negative control for antagonism or antifungal production).

If production of other antifungals is reported, additional information should be included: are these antifungals relevant for antagonism? Are their production regulated together with other antifungals? How could work the coordinated and different regulation systems? Have they an impact on antifungal production? In the case that H111 does not produce other antibiotics, what is happening with those genes present in pC3 then?

All these aspects are not clear enough to really understand the role of fragin and valdiazin, and their influence in the antagonistic phenotype. Thus, to my opinion, this aspect must be clarified and additional experimentation included, if necessary.

>>As stated above, to our knowledge the only antifungal compound produced by H111 is fragin and the only antifungal gene cluster present on pC3 is the *afc* cluster, which does not contribute to the antifungal activity of strain H111 (Agnoli et al., 2017 ; AEM 83(13), doi: 10.1128/AEM.00461-17). We have included this information in the revised version of the manuscript. The analysis of partial pC3 derivatives in this report identified a region of pC3 that is important for antifungal activity and we speculate that this portion of pC3 encodes a regulator that affects expression of the *ham* gene cluster. At present we do not know why the *afc* cluster in H111 is inactive but from our RNA-Seq data it is apparent that the *afc* genes are very poorly expressed (read counts for *afcA* is 2 compared to 31496 for *hamC*). The reason for this is not known and is beyond the scope of this manuscript.

We agree with the reviewer that Supplementary Fig. 1 benefits from complementing the quorum sensing mutant. In the original manuscript, we had used a *cepIR* double mutant, which cannot be easily complemented. We have therefore exchanged this mutant against a *cepI* mutant in the revised manuscript, which exhibits the same phenotypes as the *cepIR* double mutant. The *cepI* mutant

shows greatly reduced antifungal activity which can be restored by the exogenous addition of signal to the plate (Supplementary Fig. 1). We have also included a pC3-null derivative H111 as a control, as requested by the reviewer.

OTHER CONCERNS

Statistic is absent in all the figures. Clear differences can be seen, but statistic will strength the results.

>>We have added statistics whenever possible in the revised manuscript as requested.

Lines 104-105: I do not see the “abolishment” of the antifungal activity in Fig. 2b. It is clear that other antifungal compound still being released to the medium by the different derivative strains. It is not clearly stated if there are other antifungals released or different concentrations of one or various compounds, but a halo can be more or less observed in all the derivative mutants, but especially in Δ hamB, Δ hamD, Δ hamE and Δ hamG. An explanation would be necessary.

>> We thank the reviewer for bringing this to our attention. We thoroughly re-examined the pictures included in our manuscript as well as all other pictures that were not used for publication. However, we cannot detect a clear zone around the bacterial colony that is entirely devoid of fungal growth with any of the mutant strains. We now provide larger pictures which should allow the reader to see more clearly that there is no fungus-free zone around the bacterial colony. However, we acknowledge that the antifungal activity (no fungal growth) is somewhat blurred by differences in the pigmentation of the fungus that is observed with some of the mutant strains (slightly brown color around the bacterial colonies vs. beige/white color further away from the bacterial colony). Most likely this is a stress response of the fungus, possibly because of nutrient depletion or metabolites that are released by the different strains. We have added a short discussion regarding this phenotype in the revised manuscript.

Line 190: Change “hamD” to “ Δ hamD”; change “hamF” to “ Δ hamF”. Same in the forthcoming paged for naming mutants.

>> Corrected as requested.

Lines 367-369: is fragin a siderophore? Please, add some comments about it after your claims in these lines. Also shown in Supplementary Figures 12 and 13.

>>Fragin is shown to bind not only iron, but also other metals, particularly copper. In our manuscript, we present indirect evidence that fragin can also bind iron by showing that a high iron concentration diminishes the antifungal activity of fragin. In addition, we provide evidence, that fragin can bind copper in vitro and support this with NMR data. As fragin can bind at least iron and copper it should be classified as a metallophore. We comment on this in the discussion of the revised version of the manuscript.

Figure 1b: additional data of the protein domains of the ham genes should be included. Identity, coverage, enzyme commission numbers, etc.

>>We have expanded our bioinformatic analysis of the *ham* genes/proteins in the revised manuscript according to the requests of Reviewer #1 and #2. We now provide a more detailed analysis of the putative tailoring enzymes of the *ham* cluster (Fig. 1) as well as a comparison of % AA identity of all 60 bacterial strains that were found to have *ham* homologs (Supplementary Table 2). We are not able to present enzyme commission numbers, as the enzymes are described in the manuscript for the first time.

Figure 2: Please, correct the column H111 Δ hamF pBBR/hamF. Is repeated twice.

>>We have corrected the column description in the revised manuscript.

Figure 5: Please, clarify the role of HamB in this figure.

>>At present, we cannot assign HamB a particular function in our proposed biosynthesis pathway. As HamB is required for fragin but not valdiazin biosynthesis, we hypothesize that it acts downstream of the branching point of fragin and valdiazin biosynthesis. We have added this information in the Discussion section of the revised manuscript.

Supplementary Fig. 3. A comparison of all these ham genes could be constructed, in order to infer phylogenetic and comparative purposes among those bacterial strains. Moreover, even the analysis of the two-three genes down- and upstream the ham genes could give a valuable information about similarity among strains.

>>We agree that this figure needed improvement. We have now added a table (Supplementary Table 2) which shows the % AA identity (compared to H111) of all *ham* genes found by MultiGeneBlast analysis in a total of 60 strains of the genera *Burkholderia*, *Pandoraea* and *Pseudomonas*. In addition we have examined the genes upstream and downstream of the *ham* cluster in all these strains. We have modified Supplementary Figure 3 accordingly.

Reviewer #2 (Remarks to the Author):

In this manuscript, the authors determined the absolute configurations of two structurally related molecules, an antifungal molecule fragin and a novel signal molecule valdiazin. They demonstrated the functions as well as positive biosynthesis routes of the two molecules. Valdiazin was presented to be a new signal molecule controlling the expression of more than 150 genes, including itself and fragin. The authors also suggested the possible mode of action of the two molecules.

The major concern for the manuscript is that the premise that valdiazin is a global regulator is based on only RNA seq data (without sufficient controls) and homology in other bacteria. Valdiazin was added at a 100 μ M concentration at the start of the RNA seq experiment. This concentration has to be shown to be physiologically relevant under natural conditions of the bacterium. This concentration is also high for a compound that may have membrane perturbation activity (as possibly observed in Gram positive inhibitory activity). This nonspecific activity of membrane perturbation could be attributed to the observed RNA-seq data. Additional controls are needed in the experiments. It would be useful to isolate valdiazin from one of the fragin deficient bacteria and characterize its regulatory based activity. It is not uncommon for bio-intermediates to regulate the synthesis of end products. I think the authors have demonstrated this activity sufficiently.

>> We thank Reviewer #2 for the valuable comments. The criticism was noticed and we performed two new biological replicates of all our RNAseq experiments. Moreover, we confirmed the results of the RNA-Seq analysis by qRT-PCR of several of the identified differentially regulated genes. This novel information has been included in the manuscript and in Supplementary Table 9.

By using a quantitative HPLC approach we determined the concentration of valdiazin in the supernatant of stationary phase cultures to be approximately 22 μ M. This information is now included in the Results section. We also determined that in a *hamD* mutant background the exogenous addition of 50 μ M valdiazin to the growth medium is required to restore the promoter activities of the *hamABCDE* and *hamFG* operons to the level of the wildtype. We therefore used 50 μ M valdiazin in our new RNAseq experiments.

We have absolutely no evidence that valdiazin has membrane perturbation activity. Neither our RNA-Seq analysis nor the chemogenomic profiling in yeast provided evidence for membrane perturbation. Due to the hydrophilic nature of the molecule such an activity would also be rather unexpected. The (rather weak) inhibitory activity against Gram-positive bacteria is most likely due to the inhibition of copper-containing inner membrane proteins, which are not accessible by fragin in Gram-negative bacteria because of the outer membrane.

We have made considerable efforts to isolate valdiazin-like molecules from *P. entomophila* and *P. syringae* pv. *syringae*, which harbor the *pvf* and *mgo* cluster, respectively. So far these experiments were unsuccessful, as these molecules appear to be sufficiently different from valdiazin such that they cannot be purified by the procedures developed for valdiazin. The isolation and structure determination of these novel signals will be a study on its own.

We would like to emphasize that we do not think that valdiazin is a bio-intermediate of fragin. According to our proposed biosynthesis scheme both molecules share the initial biosynthetic steps but the final steps of their biosynthesis are different. Valdiazin is therefore not an intermediate of fragin biosynthesis but an alternative biosynthesis product derived from a subset of genes of the *ham* cluster.

Additional comments:

1. In the title, valdiazin was suggested to be a quorum sensing signal. Although the authors showed that CepIR QS mutant has around 3/7 of the bioactivity of wild type in supplementary figure 1, and *hamABCDE* and *HamFG* operons were down-regulated in cepR mutant, I do not think they explained the relationship between CepIR quorum sensing system and the *ham* cluster clearly. The reason for valdiazin to be considered as a QS signal needs to be addressed. Furthermore, global regulator and quorum sensing is different. QS would suggest that concentration is cell density dependent and no data was presented to support this.

>> At present the molecular mechanism how the CepIR QS system affects expression of the *ham* gene cluster in H111 is not known. This is currently under investigation in our lab but we believe that this information is outside of the scope of this manuscript. In this study we showed that valdiazin is a diffusible signal that autoregulates its own biosynthesis as well as that of fragin and of various other genes that we identified in our RNA-Seq analysis. We have also included novel data showing that *hamA* promoter activity is strongly induced in the late exponential phase (new Supplementary Fig. 12), indicating that production of the signal occurs at high cell densities. Taken together, valdiazin is diffusible, can induce gene expression in neighboring cells and is produced in a population density-dependent manner; these are all characteristics that suggest that valdiazin is a bona fide QS signal. We have added this information to the Discussion.

2. Figure 2a, the area of inhibition of $\Delta hamD$ is about 1/3 of wt strain, however, in line 111, the authors stated that no bioactive compound was detected in the extract of the hamD mutant. I was wondering why the hamD mutant is still active? Also in figure 2a, the last two labels in X axis are the same. I assume that the second last one should be H111 $\Delta HamF$. The HamF complement strain only has 2/3 of the activity of wt, and I wouldn't think it showed similar activity with wild type (line 102). Are the differences between the strains significant?

>>We agree that the quantified antifungal activities shown in Fig. 2 of the original manuscript appear to be in disagreement with the pictures provided in Fig. 2b. This is because of the method that we used to calculate antifungal activity in these assays, which is based on the area covered by the fungus on the agar plates. As this presentation form is misleading we decided to rather show the distance of inhibition between the bacterial colonies and the fungus as a quantitative and more commonly used measure of antifungal activity. We think that this presentation is much clearer. We also agree that the complemented *hamF* mutant shows lower activity than the wildtype strain in this assay. Importantly, however, the complemented strain shows significantly more activity than the mutant strain. Moreover, the complementation is also seen in our fungal spray assay (Fig. 2b and Supplementary Fig. 4).

3. Figure 2b, I am a little bit concerned about some of the complement strains. For example, the complement strains of $\Delta HamB$, $\Delta HamC$ and $\Delta HamG$ seem to be less active than the wt, however the authors claimed that the complemented strains showed wild type activity in line 106. I think it would be worthwhile to show the area of inhibition of all complemented strains and wt for better comparison.

>> We thank the reviewer for drawing our attention to this issue. We agree that some of the complemented strains do not show full wildtype activities. We have rephrased the sentence accordingly. We have also rearranged Fig. 2 and show larger pictures of our spray assay to increase clarity. We have also quantified these assays and show that that with none of the mutants strains antifungal activity was detectable. We have added this information as an additional figure in the Supplementary Material (Supplementary Fig. 4).

4. Supplementary figure 4, I was wondering if any peak other than fragin has any activity, based on the fact that $\Delta HamD$ has around 1/3 bioactivity as wild-type. Figure 4b, is the concentration of fragin comparable between crude extract and the HPLC-purified compound? What is the bioassay of the crude extract of *hamD* mutant looks like?

>>As stated above, we realized that the presentation of antifungal activities from our dual-culture plate assays was misleading in the original manuscript and we have changed the presentation to improve clarity in the revised manuscript. In our fungal spray assay the *hamD* mutant showed no antifungal activity. Likewise, we could not detect antifungal activity with extracts of the *hamD* mutant. Based on these observations, we concluded that the peaks present in both the wt and the *hamD* mutant extracts are unlikely to contain the antifungal molecule produced by *B. cenocepacia* H111. As there was only a single peak absent in the extract of the *hamD* mutant compared to the one of the wildtype, we only tested the fraction of the wildtype extract that contained this peak. We also tested the same fraction derived from the *hamD* mutant strain and, as expected, did not observe antifungal activity.

5. Figure 4c, the promoter activity of the hamABCDE operon was restored to wt level by supplementing 50 μM valdiazin to hamD mutant. Line 206, the authors performed RNA-seq of the wild type and the hamD mutant in the absence or presence of 100 μM valdiazin. Some explanation for the concentration choice is needed.

>> We agree that 100 μM valdiazin for the RNAseq experiment was not ideal. For the revised manuscript, we have replaced all RNAseq data from the original manuscript with a new set of two independent biological replicates, using 50 μM valdiazin (the valdiazin concentration needed to reach wt activity of the *hamABCDE* and *hamFG* promoters in a *hamD* mutant strain) for the chemical complementation. We have also performed quantitative HPLC experiments to determine the concentration of valdiazin in bacterial cultures. We calculated the concentration to be 22 μM , which is well in the concentration range we used for the chemical complementation in the revised manuscript.

6. Supplementary table 9, 3.85% of the fragin is labeled with L-leucine 13C. However, in line 301, "a 4.4% increase of 13C incorporation into fragin was observed for the leucine labeling experiment", why are the two numbers different? Line 302, the authors stated that valine is the precursor of fragin, because the incorporation of labeled valine is more than two-fold higher than that of labeled leucine. Although the incorporation rate of labeled leucine is lower than valine, the rate for leucine is not low, and leucine was predicted to be the substrate of HamD A domain (line 293). This data is not sufficient for determining substrate. First, Burkholderia are capable of utilizing just about anything as a carbon source and second, there is no data supporting that both are taken up by the bacterium at equal rates and amounts. The data as I see it, is that both are utilized by the bacterium.

>>We agree with the reviewer that these experiments are not entirely conclusive and that it is not possible to unambiguously define the fragin precursor on the basis of this experiment. For this reason we have decided to remove this set of experiments from the revised version of the manuscript.

7. Line 219, it is suggested that valdiazin activates genes involved in the metabolism of 3-methyl-2-oxobutanoate, which is the branching point between valine and leucine biosynthesis pathway. I am curious about how valdiazin affects the metabolism of 3-methyl-2-oxobutanoate. Is it facilitating the synthesis of one of the two amino acids, and is that involved in the self-regulation of the production of valdiazin?

>> Based on our new RNAseq data, we still see clear regulation of genes involved in leucine and valine biosynthesis and we do assume that this will influence fragin and valdiazin biosynthesis. Additional work will be required to elucidate the exact role of valine and leucine biosynthesis in fragin/valdiazin production, which we feel is beyond the scope of this study.

8. Line 98, no data to support it being a non-polar mutant ->

>> It is an in-frame mutant; we have clarified this in the text.

9. Line 107, need to be careful with the statement. Synthesis and transport is also important for the bacterium's antifungal activity.

>> We fully agree that synthesis and transport are important for antifungal activity. At present, we do not know how fragin is transported, but our sentence does in no way imply that transport would not be important. In fact, we discuss a possible role of a transport system that was identified among the differentially valdiazin-regulated genes in fragin transport.

10. Line 127, discussion of the 1967 paper comes after the discussion of the structure. It is out of place and should be mentioned earlier on to put your data into perspective.

>> We would not mind to mention the original paper on the identification of fragin earlier in the manuscript but we feel that we have chosen the earliest possible time point in the manuscript. We could not foresee that the *ham* cluster produces fragin before we had purified the bioactive compound and resolved its structure. In fact this came as a big surprise, as in the 1967 paper fragin was isolated from *Pseudomonas fragi*. Subsequent work, however, could not confirm fragin production by *P. fragi*.

11. Line 208, there is no discussion for only half of the genes being restored.

>> Expression of approximately 40 % of the valdiazin-regulated genes could indeed not be complemented by the external addition of synthetic valdiazin (Supplementary Table 4 and 5). Given that the $\Delta hamD$ mutant used in this experiment is unable to produce fragin, this may indicate that the observed effects on the expression of metal homeostasis genes are indirect and are caused by the metallophore activity of fragin. We have added this information to the Discussion in the revised manuscript.

12. Line 249 Fragrin ions (spelling)

>> corrected

13. Line 255-271, there are no methods describing the HIP and HOP assays.

>> The methods for the HIP-HOP analysis has been added to the revised version of the manuscript.

14. Line 312, try not to use words like surprising

>> We have removed this sentence

15. Lines 435-437, More information or reference is needed.

>> We have added additional information as requested.

16. Line 462, what was the final percentage of methanol?

>> the final methanol percentage used in the original manuscript (valdiazin concentration of 100 μ M) was 0.2 %. In the revised version of the manuscript, we have reduced the valdiazin concentration to 50 μ M and thus the methanol concentration is also reduced to 0.1

%. We have added this information in the revised manuscript.

17. Line 498, reference needed

>> We have included the missing reference

Reviewer #3 (Remarks to the Author):

This manuscript is to report the gene cluster required for the biosynthesis of the diazenium diolate compound fragin in *Burkholderia cenocepacia* strain H111. The research shows that fragin, a plant growth inhibitor, is the major antifungal compound synthesized by this bacterium. Valdiazen has been characterized to be a novel cell-to-cell signaling molecule and its biosynthesis genes have been identified in this study. The valdiazen regulon includes production of itself, and fragin biosynthesis and other >150 genes. In addition, homologs of the biosynthesis genes of valdiazen were identified in various bacteria. Seems to me that the findings may bring a wide impact on understanding gene a novel class of signal molecules. Basically, the research was hypothesis driven, the most experiments were designed logically and the manuscript were prepared nicely. The findings have expanded the knowledge on functions of fragin and could make significant impacts on studies of bacterial gene regulation.

Major questions, suggestions or comments:

1) Fragin has inhibitory actions on growth of plants, fungi and bacteria, which indicates that it has a very broad range of functions. Is the fragin activity fungicidal or fungistatic? Understanding its mode of action would beef up the research significantly. In addition, poor specificity may be a hurdler for drug development.

>>We agree that fragin has a broad range of functions, which is in agreement with our data showing that its mode of action is metal chelation and thus the compound will likely interfere with various enzymes containing metals. We also agree that poor specificity might be a hurdle for drug development. However, we did not intend to present fragin as a candidate for drug development. The intent of our manuscript is to present the first description of the biosynthetic genes for a diazeniumdiolate molecule. In the revised manuscript we stress the finding that fragin is a metallophore and that metal chelation is the molecular basis of its antifungal activity, similar to dopastin, which was demonstrated to inhibit mushroom tyrosinase that uses copper in its active center (as mentioned in the Discussion).

2) For antifungal assays, suggest using fungal spore-based methods for evaluation of antifungal activity. Typically, almost of all the strains of the *Fusarium* species easily produces conidia on common media. Spore-based method could generate more accurate quantitative results as compared agar plug. In Fig. 2b, the mutants produced different phenotypes regarding antifungal activities, which should be discussed in the manuscript. The strain names of *F. solani* (Line 387), yeast (Line 386) and bacteria (Suppl. Fig. 6) should be provided.

>> In this study we have used plates assays as well as a spray assay to assess antifungal activities, which gave reliable and comparable results. We have also tried a spore-based assay, but in our hands this did not work very well. We are convinced that changing to another assay type would not influence the conclusions drawn in our study. The strain names for yeast and *F. solani* were added as requested.

Minor questions, suggestions or comments:

1) Methods for generation of the plasmids used for genetic complementation assays should be provided as supplementary data.

>>We thank the reviewer for drawing our attention to this missing information. It has been included in the Materials section in the revised version. We also noticed that we had mislabeled our primers for *hamC* his-tagging as *hamG* expression primers in Supplementary Table 11. We have corrected this as well.

2) All scientific names of organisms in the manuscript including in the reference list should be in italic.

>>Names of organisms are now in italics throughout the text and the reference list.

3) How many biological replicates were conducted for RNA-seq results?

>>We had only used one replicate per condition for the RNAseq experiment in our original study. For the revised manuscript we have repeated our RNAseq experiments with two independent biological replicates and validated the regulation of several genes with qRT-PCR. We have also lowered the valdiazen concentration for the chemical complementation from 100 μ M to 50 μ M to be consistent with the results from our promoter fusion assays, which showed that wildtype promoter activities of the *hamABCDE* and the *hamFG* promoters required supplementation of the medium with 50 μ M valdiazen. We have replaced our original RNA-seq data in the revised manuscript against the newly generated data sets.

REVIEWERS' COMMENTS:

Reviewer #1 (Remarks to the Author):

[No further comments for author.]

Reviewer #2 (Remarks to the Author):

I am aware that the definition of a quorum sensing molecule has become muddled in recent literature. Therefore, I will not hold the authors to a standard that others are not being made to follow. The authors have responded adequately to our previous comments and I believe the manuscript is of great interest to the Burkholderia research community.

Reviewer #3 (Remarks to the Author):

[No further comments for author.]